# Adjustment Identification Distance: A `gadjid` for Causal Structure Learning

**Leonard Henckel**[1]      **Theo Würtzen**[2]      **Sebastian Weichwald**[2,3]

[1]School of Mathematics and Statistics, University College Dublin, Ireland
[2]Pioneer Centre for AI, University of Copenhagen, Denmark
[3]Department of Mathematical Sciences, University of Copenhagen, Denmark

## Abstract

Evaluating graphs learned by causal discovery algorithms is difficult: The number of edges that differ between two graphs does not reflect how the graphs differ with respect to the identifying formulas they suggest for causal effects. We introduce a framework for developing causal distances between graphs which includes the structural intervention distance for directed acyclic graphs as a special case. We use this framework to develop improved adjustment-based distances as well as extensions to completed partially directed acyclic graphs and causal orders. We develop new reachability algorithms to compute the distances efficiently and to prove their low polynomial time complexity. In our package gadjid (open source at github.com/CausalDisco/gadjid), we provide implementations of our distances; they are orders of magnitude faster with proven lower time complexity than the structural intervention distance and thereby provide a success metric for causal discovery that scales to graph sizes that were previously prohibitive.

## 1 INTRODUCTION

Inferring the causal effect of a treatment on an outcome from observational data requires qualitative knowledge of the underlying causal structure, for instance in form of a causal graph [Pearl, 2009]. We can, for example, use a causal graph to decide whether a set of covariates forms a valid adjustment set and enables correct estimation of a causal effect via adjustment [Pearl, 1995, Perkovic et al., 2018].

Under certain assumptions, we can learn the causal graph that underlies the covariates; a task known as causal discovery [e.g. Spirtes et al., 2000, Chickering, 2002, Heinze-Deml et al., 2018]. Causal discovery is challenging. First, the causal graph is only identifiable from observational data under restrictive assumptions, such as additive errors satisfying distributional or scale restrictions [Shimizu et al., 2006, Park, 2020]. Second, algorithms based on existing identifiability results often require further assumptions or approximations to be computationally feasible, for example, testing only few of the combinatorially exploding number of required conditional independence tests [Spirtes et al., 2000]. Third, causal discovery from finite data is statistically challenging and there are pitfalls to evaluating causal discovery algorithms on simulated data [Gentzel et al., 2019, Weichwald et al., 2020, Kaiser and Sipos, 2022, Reisach et al., 2021, 2023].

The literature has focused on the first two problems. Yet, for causal discovery to become practically useful, it is necessary to tackle the third problem and improve the evaluation criteria, benchmarks, and success metrics that guide algorithm development [Mooij et al., 2016, Cheng et al., 2022, Rios et al., 2021]. A prerequisite for research into more accurate causal discovery algorithms is that we quantify that accuracy, for which we need a distance between a learned graph $\mathcal{G}_{\text{guess}}$ and the true graph $\mathcal{G}_{\text{true}}$. A common and very widely used choice in the literature is the structural Hamming distance (SHD) or variants thereof which count the number of edges that differ between graphs [Tsamardinos et al., 2006, de Jongh and Druzdzel, 2009, Constantinou, 2019]. The SHD, however, does not reflect how similar graphs are when used to infer interventional distributions: A graph $\mathcal{G}_{\text{guess}}$ may have a large Hamming distance from $\mathcal{G}_{\text{true}}$ but still be a good estimate of $\mathcal{G}_{\text{true}}$ for performing causal inference (cf. Corollary 8 or [Peters and Bühlmann, 2015]). The number of edges that differ between graphs is not a performance metric for causal discovery when the graph is to be used for effect identification.

The literature on comparing causal graphs can broadly be divided into two approaches. The first approach considers data-driven graph distances [Viinikka et al., 2018, Eigenmann et al., 2020, Peyrard and West, 2021, Dhanakshirur et al., 2023]. These are challenging to use as performance

metrics for algorithm development, as evaluating these distances generally requires large samples and is only computationally feasible for small graphs. The second approach considers only the graph structure and its implications for causal inference to define a graph distance; an example is the Structural Intervention Distance (SID) [Peters and Bühlmann, 2015]. We focus on distances that consider only the graph structure. This approach has received less attention than the first approach but offers advantages: First, it enables comparisons to graphs encoding expert knowledge without having to specify all conditional distributions. Second, it is independent of the sample size, hyperparameter tuning, and choice of density estimator.

The SID counts interventional densities $f(Y \mid \mathrm{do}(T = t))$ in $\mathcal{G}_{\text{true}}$ that are incorrectly inferred if we instead use $\mathcal{G}_{\text{guess}}$ as the causal graph to compute interventional densities via parent adjustment. For directed acyclic graphs (DAGs), this amounts to counting the parent sets in $\mathcal{G}_{\text{guess}}$ that are not valid adjustment sets in $\mathcal{G}_{\text{true}}$. Using this characterization, Peters and Bühlmann [2015] provide an algorithm with $O(p^4 \log(p))$ time complexity in the number of nodes $p$.[1] They propose a generalization of the SID to completed partially directed acyclic graphs (CPDAGs), which represent equivalence classes of DAGs, by iterating over the DAGs in the equivalence class to compute a multi-set of distances. As one major use of causal graphs is inferring interventional densities, the SID is a practically relevant distance. However, parent adjustment is only one of many approaches to compute causal effects and is in fact statistically inefficient [Rotnitzky and Smucler, 2020, Henckel et al., 2022]. Further, iterating over the DAGs in a Markov equivalence class to calculate the multi-set SID between CPDAGs has exponential time complexity and the resulting non-scalar distance is difficult to interpret. In fact, the CRAN SID package v1.1 requires that the true graph be a directed acyclic graph, that is, it does not implement the distance between two completed partially directed acyclic graphs outlined by Peters and Bühlmann [2015]; calculating the SID between a DAG and CPDAG exactly likewise has exponential time complexity and returns a difficult to interpret multi-set.

**Contribution.** We develop distances between causal graphs that reflect their dissimilarity when used to infer causal effects. Specifically, we propose a framework to construct an identifiability distance from a graphical identification strategy, that is, an algorithmic approach to causal effect identification. We posit that such a distance is interesting if the underlying graphical strategy for causal effect identification represents a potential practitioner using the graph to infer causal effects. Different assumptions on how an

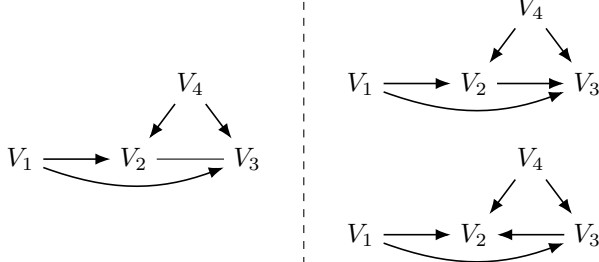

Figure 1: A CPDAG (left) and the two DAGs in the corresponding Markov equivalence class (right).

idealized practitioner would use a causal graph lead to different causal graph distances. We show that our framework includes the SID for DAGs as a special case and use the framework to propose new distances with attractive properties. We discuss for each distance a) whether the underlying identification strategy is good practice and used in practice, and b) when it is zero. Within our framework and in contrast to the SID, our distances canonically generalize to distances between any combination of DAGs and CPDAGs. We also generalize one of the distances to learned causal node orders. We develop polynomial time algorithms to compute the distances, which, to our knowledge, makes them the first causal distances between CPDAGs with a polynomial runtime guarantee. For the distances using local adjustment strategies, which in the special case of DAGs includes the SID, we show that the complexity is $O(p^2)$ for sparse and $O(p^3)$ for dense graphs, irrespective of whether the graphs are DAGs or CPDAGs. We also show that the complexity of our most complex distance is at most $O(p^4)$. We provide empirical evidence for the asymptotic time complexities and fast runtimes of our algorithms. Finally, we discuss how future advances on sound and complete criteria for causal effect identification could be integrated into our framework to develop distances for more general graph types that allow for unobserved variables.

**A `gadjid` for causal structure learning.** For our distances, we provide efficient Rust-implementations with a Python interface. Our package gadjid (open source at github.com/CausalDisco/gadjid) enables researchers to evaluate and benchmark causal discovery algorithms with causally meaningful and computationally tractable performance metrics to guide and support the development of structure learning algorithms.

## 2 PRELIMINARIES

We use graphs where nodes represent random variables and edges causal relationships. Here, we provide an overview of the key terminology and refer to Appendix A for details.

We consider two types of graphs: directed acyclic graphs (DAGs) and completed partially directed acyclic graphs

---

[1] For certain adjacency matrices, using the Strassen algorithm for matrix multiplication in the algorithm by Peters and Bühlmann [2015] may reduce the complexity to $O(p^{\log_2(7)+1} \log(p)) \approx O(p^{3.8} \log(p))$; using our novel reachability algorithms we reduce the complexity to $O(p^3)$ for dense and $O(p^2)$ for sparse graphs.

(CPDAGs), see Figure 1. DAGs are graphs with directed edges ($\rightarrow$) and without directed cycles. DAGs can describe causal relationships without feedback loops [Pearl, 2009]. They also encode conditional independences that can be read off the graph using the d-separation criterion [Pearl, 2009]. DAGs can be learned from data only under strong assumptions. However, the class of DAGs encoding the same conditional independences, known as its Markov equivalence class, can be learned under weaker assumptions. A CPDAG can uniquely represent this equivalence class if there are no hidden variables [Meek, 1995, Chickering, 2002]. CPDAGs contain directed ($\rightarrow$) and undirected (—) edges and satisfy further structural properties [Meek, 1995].

**Causal DAGs and CPDAGs.** We consider external interventions $\mathrm{do}(\mathbf{T} = \mathbf{t})$ (short $\mathrm{do}(\mathbf{t})$) for $\mathbf{T} \subseteq \mathbf{V}$ that set $\mathbf{T}$ to some value $\mathbf{t}$ for the entire population [Pearl, 1995]. A probability density function $f$ over random variables $\mathbf{V} = (V_1, \ldots, V_p)$ is compatible with a causal DAG $\mathcal{G} = (\mathbf{V}, \mathbf{E})$ if all densities $f(\mathbf{v} \mid \mathrm{do}(\mathbf{t}))$ obey

$$f(\mathbf{v} \mid \mathrm{do}(\mathbf{t})) = \begin{cases} \prod_{V \in \mathbf{V} \setminus \mathbf{T}} f(v \mid \mathrm{Pa}(V, \mathcal{G})) & \text{if } \mathbf{T} = \mathbf{t}, \\ 0 & \text{otherwise.} \end{cases}$$

This equation is known as truncated factorization formula [Pearl, 2009], manipulated density formula [Spirtes et al., 2000], or g-formula [Robins, 1986]. A density $f$ is compatible with a CPDAG $\mathcal{G}$ if it is compatible with a causal DAG in the Markov equivalence class represented by $\mathcal{G}$.

**Identifying formula.** Causal graphs are used to estimate the causal effect of a treatment $\mathbf{T} \subseteq \mathbf{V}$ on an outcome $\mathbf{Y} \subseteq \mathbf{V}$ from observational data, that is, to estimate (functionals of) the interventional distribution $f(\mathbf{y} \mid \mathrm{do}(\mathbf{t}))$. To do so, we require an identifying formula for this interventional distribution, that is, an equation in the observational density that solves for $f(\mathbf{y} \mid \mathrm{do}(\mathbf{t}))$ for any $f$ compatible with the causal graph. We refer to inferring such an identifying formula from a causal graph as inferring the causal effect. An effect is identifiable in a causal graph $\mathcal{G}$ if there is at least one identifying formula.

**Valid adjustment.** Let $\mathbf{T}, \mathbf{Y}$, and $\mathbf{Z}$ be pairwise disjoint node sets in a causal DAG or CPDAG $\mathcal{G}$. $\mathbf{Z}$ is a valid adjustment set if $f(\mathbf{y} \mid \mathrm{do}(\mathbf{t})) = \int f(\mathbf{y} \mid \mathbf{t}, \mathbf{z}) f(\mathbf{z}) \, \mathrm{d}\mathbf{z}$ for any density $f$ compatible with $\mathcal{G}$. Graphical criteria fully characterize valid adjustment sets in DAGs, CPDAGs, and other graph types [Perkovic et al., 2018].

**Causal ordering.** Let $\mathcal{G}$ be a DAG with node set $\mathbf{V}$. A strict partial order $\prec$ on $\mathbf{V}$ is called a causal order of $\mathcal{G}$ if for all nodes $A, B \in \mathbf{V}$ with $A \rightarrow B$ it holds that $A \prec B$. In general, there are multiple causal orders. For two DAGs $\mathcal{G}$ and $\mathcal{H}$ with node set $\mathbf{V}$, we say that $\mathcal{G}$ respects the causal orders of $\mathcal{H}$ if every causal order of $\mathcal{G}$ is a causal order of $\mathcal{H}$. We define $\mathrm{pre}_\prec(B) = \{A \in \mathbf{V} \mid A \prec B\}$, $\mathrm{post}_\prec(A) =$

$\{B \in \mathbf{V} \mid A \prec B\}$, and $\mathcal{G}_\prec$ as the transitively closed DAG with $A \rightarrow B$ if and only if $A \prec B$.

# 3 CAUSAL IDENTIFICATION DISTANCE

We introduce a framework for developing identifiability distances between causal graphs. This framework lays out how to extend distances to different graph types and align them with how causal graphs are used to answer causal queries.

## 3.1 FRAMEWORK

In our framework, a distance is defined by a) a sound and complete identification strategy and b) a verifier. We use the identification strategy to derive identification formulas based on $\mathcal{G}_{\mathrm{guess}}$ and the verifier to evaluate whether the identification formulas obtained on $\mathcal{G}_{\mathrm{guess}}$ are correct in $\mathcal{G}_{\mathrm{true}}$. Intuitively, the identification strategy represents how an idealized practitioner would use $\mathcal{G}_{\mathrm{guess}}$ to infer causal effects and the verifier evaluates how often the practitioner would be wrong if the ground truth graph were $\mathcal{G}_{\mathrm{true}}$. For simplicity, we only consider single-node interventions while the framework generalizes when provided a sound and complete identification strategy and verifier (cf. also Section 4.3).

**Definition 1** *(Identification Strategy).* An *identification strategy* is an algorithm that for a tuple $(\mathcal{G}, T, Y)$ of a causal graph $\mathcal{G}$ and two distinct nodes $T$ and $Y$ in $\mathcal{G}$, returns the tuple $(T, Y)$ and either an identifying formula $I$ for $f(y \mid \mathrm{do}(t))$ or none. An identification strategy $\mathcal{I}$ is sound and complete if a) $\mathcal{I}(\mathcal{G}, T, Y) \neq$ none if and only if $f(y \mid \mathrm{do}(t))$ is identifiable in $\mathcal{G}$ and b) all returned identifying formulas are correct for any $f$ compatible with $\mathcal{G}$.

**Example 2** *(Parent Adjustment Strategy).* For a DAG $\mathcal{G}$ and two distinct nodes $T$ and $Y$, $\mathbf{P}_T = \mathrm{Pa}(T, \mathcal{G})$ is a valid adjustment set whenever $Y \notin \mathbf{P}_T$. If, on the other hand, $Y \in \mathbf{P}_T$, then, by the acyclicity of $\mathcal{G}$, $Y$ is a non-descendant of $T$ and so there is no causal effect from $T$ on $Y$. We can combine these two results to obtain the sound and complete *parent adjustment strategy*

$$\mathcal{I}_P(\mathcal{G}, T, Y) = \begin{cases} \int f(y \mid t, \mathbf{p}_T) f(\mathbf{p}_T) \, \mathrm{d}\mathbf{p}_T & \text{if } Y \notin \mathbf{P}_T, \\ f(y) & \text{else.} \end{cases}$$

In a DAG, all causal effects are identifiable and therefore $\mathcal{I}_P$ never returns none.

**Definition 3** *(Verifier).* A *verifier* is an algorithm that given a graph $\mathcal{G}$, distinct nodes $T$ and $Y$ in $\mathcal{G}$, and an identifying formula $I$ for $f(y \mid \mathrm{do}(t))$, verifies whether $I$ is correct for all densities compatible with $\mathcal{G}$ and returns either correct or incorrect. For an input of none it verifies that the effect is not identifiable in $\mathcal{G}$, that is, that no identifying formula exists.

Identification has been widely studied; for example, various sufficient conditions for the validity of adjustment sets are known [Pearl, 1993, Maathuis and Colombo, 2015]. Verification, however, has received limited attention and no algorithm is available to verify an arbitrary identifying formula for DAGs or CPDAGs. Yet, for some types of identifying formulas, necessary and sufficient graphical criteria exist and these can be used for verification. In particular, for identifying formulas that use adjustment, we can use the necessary and sufficient adjustment criterion for verification [Shpitser et al., 2010, Perkovic et al., 2018]. A necessary and sufficient criterion also exists for instrumental variables but only for linear models [Henckel et al., 2023]. We focus on adjustment-based identification strategies but our framework is amenable to other strategies provided a corresponding verifier exists.

**Example 4** (*Adjustment-Verifier for DAGs*). The identification strategy $\mathcal{I}_P$ relies on two identification principles: a) valid adjustment and b) non-descent. The verifier $\mathcal{V}_{\text{adj}}$ in Algorithm 1 is simple, sound, and complete and uses the adjustment criterion and a non-descent check for the verification of adjustment-based identification strategies. In DAGs all effects are identifiable and so the verifier rejects any none-identification formula as incorrect, that is, the if-branch in line 5 is not reached and only included for completeness.

**Definition 5** (*$\mathcal{I}$-Specific Identification Distance*). Given a sound and complete identification strategy $\mathcal{I}$, we define the *$\mathcal{I}$-specific identification distance* between two graphs $\mathcal{G}_{\text{true}}$ and $\mathcal{G}_{\text{guess}}$ with common node set $\mathbf{V}$ as the number of identification formulas inferred by $\mathcal{I}$ on $\mathcal{G}_{\text{guess}}$ that are incorrect relative to $\mathcal{G}_{\text{true}}$, that is,

$$d^{\mathcal{I}}(\mathcal{G}_{\text{true}}, \mathcal{G}_{\text{guess}}, \mathbf{S})$$
$$= \sum_{(T,Y) \in \mathbf{S}} \mathbb{1}_{\{\text{incorrect}\}}\big(\mathcal{V}(\mathcal{G}_{\text{true}}, \mathcal{I}(\mathcal{G}_{\text{guess}}, T, Y))\big)$$

where $\mathbf{S} \subseteq \overline{\mathbf{S}} = \{(T,Y) \in \mathbf{V} \times \mathbf{V} \mid T \neq Y\}$ and $\mathcal{V}$ is a verifier. Unless otherwise noted, we use all $p(p-1)$ pairs of distinct nodes in $\mathbf{V}$ and write $d^{\mathcal{I}}(\mathcal{G}_{\text{true}}, \mathcal{G}_{\text{guess}}) = d^{\mathcal{I}}(\mathcal{G}_{\text{true}}, \mathcal{G}_{\text{guess}}, \overline{\mathbf{S}})$.[2]

Formally, the $\mathcal{I}$-specific identification distance between $\mathcal{G}_{\text{true}}$ and $\mathcal{G}_{\text{guess}}$ counts how many of the identification formulas obtained by using the identification strategy $\mathcal{I}$ on $\mathcal{G}_{\text{guess}}$ are incorrect for $\mathcal{G}_{\text{true}}$. Intuitively, the distance is the number of causal effects an idealized practitioner would wrongly infer from $\mathcal{G}_{\text{guess}}$, if $\mathcal{I}$ resembled how the practitioner would use the graph $\mathcal{G}_{\text{guess}}$ for causal inference while the true DAG

---

[2]The flexibility to choose other sets $\mathbf{S}' \subset \overline{\mathbf{S}}$ allows one to tailor the distance $d^{\mathcal{I}}(\mathcal{G}_{\text{true}}, \mathcal{G}_{\text{guess}}, \mathbf{S}')$ to consider only some specific nodes of interest as treatment or effect nodes or, given a suitable identification strategy and verifier, to consider multi-node interventions when comparing graphs (see also Section 4.3).

---

**Algorithm 1** Adjustment-Verifier $\mathcal{V}_{\text{adj}}$
1: **Input**: Graph $\mathcal{G}$, tuple $(T, Y)$, identifying formula $I$
2: **Output**: Validity indicator $V \in \{\text{correct}, \text{incorrect}\}$
3: $V \leftarrow \text{incorrect}$
4: **if** $I = \text{none}$ and $f(y \mid \text{do}(t))$ not identifiable in $\mathcal{G}$ **then**
5:    $V \leftarrow \text{correct}$
6: **else if** $I = f(y)$ and $Y \in \text{NonDe}(T, \mathcal{G})$ **then**
7:    $V \leftarrow \text{correct}$
8: **else if** $I = \int f(y \mid t, \mathbf{z})f(\mathbf{z}) \ d\mathbf{z}$ and $\mathbf{Z}$ is a valid adjustment set relative to $(T, Y)$ in $\mathcal{G}$ **then**
9:    $V \leftarrow \text{correct}$
10: **return** V

---

were $\mathcal{G}_{\text{true}}$. By construction, strategy-specific identification distances are asymmetric in their input graphs.

**Example 6** (*SID is the $\mathcal{I}_P$-Specific Distance for DAGs*). Let $\mathcal{G}_{\text{true}}$ and $\mathcal{G}_{\text{guess}}$ be DAGs with common node set $\mathbf{V}$. Then $d^{\mathcal{I}_P}(\mathcal{G}_{\text{true}}, \mathcal{G}_{\text{guess}})$ coincides with $\text{SID}(\mathcal{G}_{\text{true}}, \mathcal{G}_{\text{guess}})$ as defined by Peters and Bühlmann [2015]. For CPDAGs, however, the SID is defined as the multi-set obtained by calculating the SID for each DAG in the Markov equivalence class and is not a distance in our framework; in Section 5 we present the canonical extension of the $\mathcal{I}_P$-specific distance to CPDAGs that outputs a scalar and retains interpretability.

## 3.2 GENERAL PROPERTIES

Any identification distance between a DAG and a supergraph of that DAG is zero. A corollary highlights that identification distances differ from the SHD: there exist DAGs for which identification distances are maximally different from the SHD. Proofs are provided in Appendix B.

**Proposition 7** (Distance to Super-DAG is Zero). *Let $\mathcal{I}$ be a sound and complete identification strategy for DAGs. For any DAG $\mathcal{G}_{\text{true}}$, it holds that if $\mathcal{G}_{\text{guess}}$ is a super-DAG of $\mathcal{G}_{\text{true}}$, then $d^{\mathcal{I}}(\mathcal{G}_{\text{true}}, \mathcal{G}_{\text{guess}}) = 0$.*

As $d^{\mathcal{I}}(\mathcal{G}_{\text{true}}, \mathcal{G}_{\text{true}}) = 0$, identification distances are pre-metrics. Proposition 7 is a consequence of all causal effects in a DAG being identifiable; adding edges removes the information that certain effects are absent and may reduce the number of correct identifying formulas, but never to zero. For other graph types such as CPDAGs, this is not the case (cf. Section 5). As a corollary, DAGs may be close in identification distance but far in SHD.

**Corollary 8** (Identification Distances Differ from the SHD). *Let $d^{\mathcal{I}}$ be a strategy-specific identification distance. Let $\mathcal{G}_{\text{guess}}$ be a fully connected DAG with $p$ nodes and $\mathcal{G}_{\text{true}}$ the empty DAG on the same node set. Then the SHD $d_H$ is maximal and maximally different from $d^{\mathcal{I}}$:*

$$d_H(\mathcal{G}_{\text{true}}, \mathcal{G}_{\text{guess}}) - d^{\mathcal{I}}(\mathcal{G}_{\text{true}}, \mathcal{G}_{\text{guess}}) = p(p-1)/2 - 0.$$

# 4 DAG DISTANCES

We propose three adjustment-based distances for DAGs and extend them to CPDAGs in Section 5. We propose a *parent adjustment distance* which between DAGs corresponds to the SID but, in contrast to the SID, generalizes canonically to CPDAGs. We develop an *ancestor adjustment distance* that assigns low distance to graphs with similar causal orders and an *Oset adjustment distance* that uses a statistically efficient identification strategy. We discuss how each distance corresponds to different assumptions on how a practitioner would use a graph for causal inference. Users need to choose (a combination of) distances based on how they envision the graph will be used for causal reasoning in a downstream task. Depending on the downstream task, even the SHD may be considered a causal graph distance, for example, if the downstream task merely involves reasoning about the existence of direct cause-effect relationships but not the identification or estimation of those direct effects.

## 4.1 PARENT ADJUSTMENT DISTANCE

We call $d^{\mathcal{I}_P}$ the *parent adjustment distance (Parent-AID)*. The Parent-AID is an identification distance (cf. Example 6) but yields unintuitive results between graphs with the same causal orders (cf. Lemma 10) and uses an inefficient adjustment strategy (cf. Section 4.3); we include it for completeness, as it includes the SID for DAGs as a special case, but, in contrast to the SID, canonically extends to CPDAGs within our framework (cf. Section 5). We develop refined adjustment-based distances in the next two subsections.

Parent adjustment is used in practice [Gascon et al., 2015, Sunyer et al., 2015]. To reason about the effect of intervening on a variable $T$, it requires one to know only the direct causes of $T$ but not the full causal graph; if a practitioner knew the parent sets for all nodes infact they would know the full causal DAG. The Parent-AID assumes a practitioner who follows this common practice. The parent adjustment strategy is local, that is, for any pair $(T, Y)$ the adjustment set only depends on $T$ but not $Y$. We use this to improve the time complexity of calculating the distance (cf. Section 6). For DAGs, $d^{\mathcal{I}_P}(\mathcal{G}_{\text{true}}, \mathcal{G}_{\text{guess}}) = 0$ if and only if $\mathcal{G}_{\text{guess}}$ is a supergraph of $\mathcal{G}_{\text{true}}$ [Peters and Bühlmann, 2015].

## 4.2 ANCESTOR ADJUSTMENT DISTANCE

Many causal discovery algorithms learn an ordering of the nodes in a separate first step [Shojaie and Michailidis, 2010, Bühlmann et al., 2014, Chen et al., 2019, Park, 2020] and pairwise causal effects can be learned from the correct causal order alone [Bühlmann et al., 2014, Section 2.6]. We formalize this known result as follows.

**Lemma 9** (Ancestors are Valid Adjustment Sets). *Let $T$ and $Y$ be two distinct nodes in a DAG $\mathcal{G}$. Then any set*

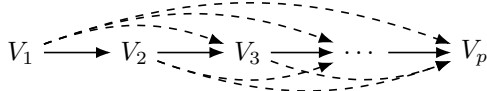

Figure 2: Fully connected and chain DAG in Lemma 10.

$\mathbf{Z}$ *such that* $\mathrm{Pa}(T, \mathcal{G}) \subseteq \mathbf{Z} \subseteq \mathrm{NonDe}(T, \mathcal{G})$ *and* $Y \notin \mathbf{Z}$ *is a valid adjustment set for* $(T, Y)$ *in* $\mathcal{G}$. *As a corollary, given an order* $\prec$, $\mathrm{pre}_{\prec}(T)$ *is a valid adjustment set for all* $Y \notin \mathrm{pre}_{\prec}(T)$ *in all DAGs for which* $\prec$ *is a causal order.*

Thus, it is possible to derive identification formulas in $\mathcal{G}_{\text{guess}}$ that are also correct in $\mathcal{G}_{\text{true}}$, if $\mathcal{G}_{\text{guess}}$ respects the causal orders of $\mathcal{G}_{\text{true}}$. Yet, even if two DAGs have the same causal orders, the Parent-AID between them can be large.

**Lemma 10** (Parent-AID Misrepresents Causal Order). *Let* $\mathcal{G}_{\text{true}}^p$ *be the fully connected DAG over* $p$ *causally-ordered nodes* $\{V_1, ..., V_p\}$ *and* $\mathcal{G}_{\text{guess}}^p$ *the chain* $V_1 \to V_2 \cdots \to V_p$ *(cf. Figure 2). Then, despite $\mathcal{G}_{\text{true}}$ and $\mathcal{G}_{\text{guess}}$ respecting each others causal orders,* $d^{\mathcal{I}_P}(\mathcal{G}_{\text{true}}^p, \mathcal{G}_{\text{guess}}^p) = p^2 - 4p + 4$ *which is close to its maximal value* $p(p-1)$ *in the sense that*

$$\lim_{p \to \infty} d^{\mathcal{I}_P}(\mathcal{G}_{\text{true}}^p, \mathcal{G}_{\text{guess}}^p)/p(p-1) = 1.$$

Considering DAGs as distant that respect each others causal orders may be unintuitive. We therefore propose an alternative adjustment strategy and distance.

**Definition 11** (*Ancestor Adjustment Strategy and Distance*). Given two distinct nodes $T$ and $Y$ in a DAG $\mathcal{G}$, let $\mathbf{A}_T = \mathrm{An}(T, \mathcal{G}) \setminus \{T\}$ and $\mathbf{D}_T = \mathrm{De}(T, \mathcal{G})$. We define the *ancestor adjustment strategy* as

$$\mathcal{I}_A(\mathcal{G}, T, Y) = \begin{cases} \int f(y \mid t, \mathbf{a}_T) f(\mathbf{a}_T) \, \mathrm{d}\mathbf{a}_T & \text{if } Y \in \mathbf{D}_T, \\ f(y) & \text{else,} \end{cases}$$

which is a sound and complete identification strategy per Proposition 12. We call the corresponding distance $d^{\mathcal{I}_A}$ the *ancestor adjustment distance (Ancestor-AID)*.

**Proposition 12** (Ancestor-AID Reflects Causal Order). *The ancestor adjustment strategy $\mathcal{I}_A$ is sound and complete for DAGs and so the corresponding ancestor adjustment distance $d^{\mathcal{I}_A}$ is the $\mathcal{I}_A$-specific identification distance. Further, for any two DAGs $\mathcal{G}_{\text{true}}$ and $\mathcal{G}_{\text{guess}}$ with the same node set, $d^{\mathcal{I}_A}(\mathcal{G}_{\text{true}}, \mathcal{G}_{\text{guess}}) = 0$ if and only if $\mathcal{G}_{\text{guess}}$ respects the causal orders of $\mathcal{G}_{\text{true}}$.*

Due to Proposition 12, the Ancestor-AID is preferable to the Parent-AID for evaluating the causal order of a learned graph. It can be used both as a replacement and to complement the Parent-AID.

The ancestor adjustment strategy is local, that is, for any pair $(T, Y)$ the adjustment set only depends on $T$ but not

$Y$. Adjusting for the ancestors has some advocates [Rubin, 2008] and is at least as statistically efficient as parent adjustment [Henckel et al., 2022]. A practitioner more confident in the ability of causal discovery algorithms to learn causal orders rather than all specific edges and exact parent sets, may prefer the ancestor adjustment strategy over the parent adjustment strategy to infer causal effects from a learned graph.

## 4.3 OSET ADJUSTMENT DISTANCE

In practice, identifying formulas are a tool to estimate a causal effect of interest. Different identifying formulas correspond to different estimators for this effect. For example, in linear models we can estimate the average treatment effect with an ordinary least squares regression of $Y$ on $T$ and $\mathbf{Z}$; this estimator is consistent for any valid adjustment set $\mathbf{Z}$. Other properties of the estimator, such as its asymptotic variance, however, depend on the adjustment set.

We can use the causal graph to decide which valid adjustment sets result in statistically efficient estimators; importantly, for a large class of estimators, the valid adjustment set $\mathrm{Pa}(T, \mathcal{G})$ is close to the least efficient among all valid adjustment sets [Rotnitzky and Smucler, 2020, Witte et al., 2020, Henckel et al., 2022]. The parents are therefore an inefficient adjustment set and parent adjustment is perhaps not good practice. As an alternative Henckel et al. [2022] have proposed the optimal adjustment set.

**Definition 13** *(Optimal Adjustment Set (Oset))*. Let $T$ and $Y$ be two distinct nodes in a DAG $\mathcal{G}$. Then the *optimal adjustment set (Oset)* $\mathbf{O}(T, Y, \mathcal{G})$ is defined as

$$\mathbf{O}(T, Y, \mathcal{G}) = \mathrm{Pa}(\mathrm{Cn}(T, Y, \mathcal{G}), \mathcal{G}) \setminus \mathrm{Forb}(T, Y, \mathcal{G})$$

where $\mathrm{Cn}(T, Y, \mathcal{G})$ are the causal and $\mathrm{Forb}(T, Y, \mathcal{G})$ the forbidden nodes as defined in Appendix A.

If $Y \in \mathrm{De}(T, \mathcal{G})$, then $\mathbf{O}(T, Y, \mathcal{G})$ is a valid adjustment set whenever a valid adjustment set exists. For a large class of estimators, the Oset is the most statistically efficient among all valid adjustment sets. We use this result to propose another adjustment-based identification distance.

**Definition 14** *(Oset Adjustment Strategy and Distance)*. Given two distinct nodes $T$ and $Y$ in a DAG $\mathcal{G}$, let $\mathbf{O}_T = \mathbf{O}(T, Y, \mathcal{G})$ and $\mathbf{D}_T = \mathrm{De}(T, \mathcal{G})$. We define the *Oset adjustment strategy* as

$$\mathcal{I}_O(\mathcal{G}, T, Y) = \begin{cases} \int f(y \mid t, \mathbf{o}_T) f(\mathbf{o}_T) \, \mathrm{d}\mathbf{o}_T & \text{if } Y \in \mathbf{D}_T, \\ f(y) & \text{else,} \end{cases}$$

which is a sound and complete identification strategy per Proposition 15. We call the corresponding distance $d^{\mathcal{I}_O}$ the *Oset adjustment distance (Oset-AID)*.

**Proposition 15** (Oset-AID is the $\mathcal{I}_O$-Specific Distance). *The Oset adjustment strategy $\mathcal{I}_O$ is sound and complete for DAGs and so the corresponding Oset adjustment distance $d^{\mathcal{I}_O}$ is the $\mathcal{I}_O$-specific identification distance.*

Oset adjustment has seen some early adoption by practitioners [Steiger et al., 2021]. Given the Oset's efficiency guarantee, the Oset-AID assumes that a practitioner takes efficiency into account when selecting valid adjustment sets. The Oset adjustment strategy is non-local as the Oset depends on both $T$ and $Y$. As a result, the Oset adjustment distance is computationally expensive with polynomial complexity of one order higher than that of the Parent- and Ancestor-AID which use local adjustment strategies (cf. Section 6). Further, we do not have a graphical characterization of all cases where the Oset adjustment distance is zero (cf. Example 18 in Appendix C).

**Joint interventions.** Another advantage of the Oset adjustment strategy over the parent or ancestor adjustment strategies is that the Oset—in contrast to the parents or ancestors— is a valid adjustment set whenever a valid adjustment set exists, even if we consider joint interventions. As such, the Oset adjustment strategy enables generalizations of the Oset adjustment distance to settings where $\mathbf{S}$ may contain multinode interventions. However, adjustment is not sound and complete for effects of joint interventions and there may exist identifiable effects that are not identifiable via adjustment. Therefore, this generalization is strictly speaking not a strategy-specific identification distance. Identifiable joint intervention effects that cannot be identified via adjustment are characterized in Corollary 27 of Perkovic et al. [2018] and can be correctly identified by other strategies [Nandy et al., 2017, Huang and Valtorta, 2006]. However, there is no verifier for these alternative strategies. More research into verification is required to develop a proper strategy-specific distance that considers joint interventions.

## 5 CPDAG DISTANCES

In general and without strong assumptions, the true causal DAG cannot be identified or learned, even from infinite data [e.g. Peters et al., 2014]. Instead, many causal discovery algorithms target the corresponding Markov equivalence class and aim to learn its CPDAG [e.g. Chickering, 2002]. To evaluate common causal discovery algorithms, we thus also need easy to compute and interpret distances between CPDAGs. Our strategy-specific distance framework provides a recipe on how to develop such distances: a) devise a sound and complete identification strategy and b) devise a corresponding verifier. We follow this recipe to propose new and computationally attractive distances for CPDAGs based on the parent, ancestor, and Oset adjustment strategies. The distances operate directly on the CPDAGs, assess the compatibility of identification formulas between the two graphs,

and return a scalar distance. This improves upon previous approaches that iterate over an exponentially large number of Markov equivalent DAGs to calculate a multi-set of distances, which is computationally prohibitive and difficult to interpret. We also discuss potential distances across graph types and their pitfalls.

## 5.1 CPDAG TO CPDAG DISTANCES

In contrast to DAGs, not all causal effects are identifiable given a CPDAG [Meek, 1995]. For example, we cannot identify the effect of $A$ on $B$ given the CPDAG $A - B$. To extend the SID to CPDAGs, Peters and Bühlmann [2015] combine two approaches. The first, is to consider all DAGs in the Markov equivalence class of the CPDAG $\mathcal{G}_{\text{guess}}$ and compute a multi-set of distances using the SID for DAGs. The second, is to ignore all $(T, Y)$ node tuples for which the effect of $T$ on $Y$ is not identifiable in $\mathcal{G}_{\text{true}}$. In principle, the first approach could also be applied to $\mathcal{G}_{\text{true}}$ and the second to $\mathcal{G}_{\text{guess}}$. Either way, both approaches of extending the SID to CPDAGs have drawbacks.

The first approach yields a difficult to interpret multi-set of values and is in general computationally infeasible beyond graphs with very few nodes and small Markov equivalence class. For example, let $\mathcal{G}_{\text{true}}$ be the empty CPDAG (which is also a DAG) and $\mathcal{G}_{\text{guess}}$ the fully connected CPDAG. The SID between the empty and the fully connected CPDAG is a multi-set of 0s but to compute it one needs to iterate over all fully connected DAGs. Arguably, the two CPDAGs are also maximally different when used to infer causal effects, since all effects are identifiable in the empty CPDAG but no effect is identifiable in the fully connected CPDAG. The second approach discards valuable information. For example, if no effect is identifiable in $\mathcal{G}_{\text{true}}$, then the SID between $\mathcal{G}_{\text{true}}$ and any other CPDAG $\mathcal{G}_{\text{guess}}$ is a multi-set of 0s. The SID between CPDAGs as proposed by Peters and Bühlmann [2015] inherits these drawbacks[3] and in contrast to the SID between DAGs, is not an easy-to-interpret identification distance within our framework.

Our framework offers an alternative. Indeed, following our framework of identification distances there is a canonical solution to developing distances between CPDAGs: have identification strategies return none in case an effect is non-identifiable in $\mathcal{G}_{\text{guess}}$ and treat non-identifiability as a claim that we can verify in $\mathcal{G}_{\text{true}}$ just like we verify identifying formulas returned by the identification strategy. Identification distances always return a single scalar value and – by sidestepping iteration over exponentially large Markov equivalence classes and using efficient verification algorithms on CPDAGs – they are computationally tractable even for large CPDAGs. Furthermore, as identification dis-

tances they are interpretable since they capture how often a practitioner would wrongly infer a causal effect when using the learned instead of the true CPDAG. Importantly, we do not presuppose that a practitioner would (randomly) pick a DAG within the Markov equivalence class of the learned CPDAG and just use that DAG for causal inference; instead, we posit a practitioner would only infer those effects that are identifiable in the learned CPDAG and would rather look into learning a more refined graph than attempting to reason about non-identifiable effects.

As a result, however, there is no clear relationship of the strategy-specific identification distances between two DAGs, to the distances between the two corresponding CPDAGs. For example, the DAGs $V_1 \rightarrow \cdots \rightarrow V_p$ and $V_1 \leftarrow \cdots \leftarrow V_p$ have the same CPDAG and yet the adjustment distance between the two DAGs is maximal (irrespective of the strategy). In contrast to DAGs (cf. Section 3.2), adding edges in CPDAGs may render some causal effects non-identifiable and no analogous statement to Proposition 7 holds for identification distances between CPDAGs; indeed, since all effects are identifiable in the empty CPDAG but none in the fully connected CPDAG, their identification distance is maximal. Further, the distance between any two CPDAGs in which no effect is identifiable is zero. For example, if $\mathcal{G}_{\text{true}}$ and $\mathcal{G}_{\text{guess}}$ are both CPDAGs consisting of a single undirected path connecting all nodes, then their distance is zero even though they may not have a single edge in common. While this behavior is less extreme than for the SID where the distance between a true CPDAG with no identifiable effects and any learned CPDAG is zero, it may nonetheless seem unintutive. However, following the interpretation of identification distances, the distance of zero between any two CPDAGs in which no effect is identifiable is reasonable since a practitioner given such a $\mathcal{G}_{\text{guess}}$ would conclude that no effect is identifiable as is indeed the case in $\mathcal{G}_{\text{true}}$.

In CPDAGs, non-identifiability is characterized by a graphical condition called amenability by Perkovic et al. [2018], Perkovic [2020], which for single-node interventions is as follows.

**Proposition 16** (Amenability). *Consider distinct nodes $T$ and $Y$ in a CPDAG $\mathcal{G}$. The interventional density $f(y \mid \mathrm{do}(t))$ is identifiable if and only if there exists no possibly directed path from $T$ to $Y$ that starts with an undirected edge. If this holds, we say that $(\mathcal{G}, T, Y)$ is amenable.*

Equipped with graphical conditions for non-identifiability and validity of adjustment sets in CPDAGs, we can apply the adjustment verifier in Algorithm 1 to CPDAGs. Within our framework, any sound and complete adjustment-based identification strategy together with this verifier defines a strategy-specific identification distance for CPDAGs. To extend the above adjustment distances to CPDAGs, we need to extend the identification strategies by adding an amenability

---

[3]In fact, the CRAN SID package v1.1 requires the true graph to be a DAG and does not implement the distance between two CPDAGs outlined by Peters and Bühlmann [2015].

check such that none is returned for $(\mathcal{G}, T, Y)$ that are not amenable and else the return values of the identification strategies $\mathcal{I}_P$, $\mathcal{I}_A$, and $\mathcal{I}_O$ are the same as for DAGs.

The parent adjustment strategy is sound and complete for CPDAGs [Maathuis and Colombo, 2015, Corollary 4.2], as are the Oset [Henckel et al., 2022, Theorem 3] and ancestor adjustment strategies (Proposition 20, Appendix B). Thus, we generalize the Parent-AID, the Ancestor-AID, and the Oset-AID to distances between CPDAGs.

## 5.2   DAG, CPDAG, AND ORDER DISTANCES

**DAG to CPDAG distance.**   Given a suitable identification strategy and verifier, we can define a strategy-specific distance between graphs of different type. For example, since the presented adjustment strategies and verifier apply to both DAGs and CPDAGs, our distances accept any combination of DAG and CPDAG as $\mathcal{G}_{\text{true}}$ and $\mathcal{G}_{\text{guess}}$. Yet, such a distance may be unintuitive: The distance between a DAG $\mathcal{G}_{\text{true}}$ and the CPDAG $\mathcal{G}_{\text{guess}}$ that encodes the Markov equivalence class of $\mathcal{G}_{\text{true}}$ is generally non-zero as some effects are non-identifiable in the CPDAG; the distance to this correct CPDAG may in fact be further than to another CPDAG that encodes a Markov equivalence class that does not contain $\mathcal{G}_{\text{true}}$ (cf. Example 19, Appendix C). A cross-graph-type distance may still be useful when comparing an algorithm that can learn a DAG, such as LiNGAM [Shimizu et al., 2006], to an algorithm that cannot, such as GES [Chickering, 2002]. Our implementation therefore accepts DAG-to-CPDAG and CPDAG-to-DAG comparisons.

**Transformations to compare alike.**   An alternative is to transform one graph type to the other and then apply a distance between DAGs or between CPDAGs. To obtain a proxy for the distance between a DAG $\mathcal{G}_{\text{true}}$ and a CPDAG $\mathcal{G}_{\text{guess}}$, for example, one could pick a DAG corresponding to $\mathcal{G}_{\text{guess}}$ and compare that to $\mathcal{G}_{\text{true}}$; common approaches are a) to sample a DAG in the Markov equivalence class of $\mathcal{G}_{\text{guess}}$ or b) to orient undirected edges in $\mathcal{G}_{\text{guess}}$ for which a corresponding edge in DAG $\mathcal{G}_{\text{true}}$ exists correctly and the remaining undirected edges randomly while ensuring acyclicity. Both approaches are ad-hoc, non-deterministic, and ignore causal information in the CPDAG $\mathcal{G}_{\text{guess}}$, such as claims about which effects are not identifiable. Our CPDAG distance enables a principled alternative: transform the DAG $\mathcal{G}_{\text{true}}$ to its corresponding CPDAG and then compare the CPDAG corresponding to the true DAG to the learned CPDAG. This approach is natural, when the test data is simulated according to a DAG $\mathcal{G}_{\text{true}}$ and we compare the performance of two CPDAG learning algorithms, such as GES and PCALG[4] [Chickering, 2002, Spirtes et al., 2000].

---

[4]In finite samples the output of PCALG may not be a CPDAG and for these graphs no identification strategy exists. As a result, it may be necessary to resolve PCALG conflicts or to use non-causal

**DAG to order distance.**   Given a strict partial order $\prec$ on nodes $\mathbf{V}$ we can define the identification strategy

$$
\mathcal{I}_{\text{ord}}(\prec, T, Y) = \begin{cases} \int f(y \mid t, \mathbf{b}_T) f(\mathbf{b}_T) \, \mathrm{d}\mathbf{b}_T & \text{if } Y \in \mathbf{A}_T, \\ f(y) & \text{else,} \end{cases}
$$

where $\mathbf{B}_T = \text{pre}_{\prec}(T)$ and $\mathbf{A}_T = \text{post}_{\prec}(T)$. By Lemma 9 this strategy is sound and complete and we can verify the returned identification formulas in a DAG $\mathcal{G}_{\text{true}}$ using $\mathcal{V}_{\text{adj}}$. As such we obtain a strategy-specific distance $d^{\mathcal{I}_{\text{ord}}}(\mathcal{G}_{\text{true}}, \prec_{\text{guess}}) = d^{\mathcal{I}_A}(\mathcal{G}_{\text{true}}, \mathcal{G}_{\prec_{\text{guess}}})$ between DAGs and strict partial orders. The strategy-specific distance $d^{\mathcal{I}_{\text{ord}}}(\mathcal{G}_{\text{true}}, \prec_{\text{guess}})$ counts the number of identification formulas derived from the partial order $\prec_{\text{guess}}$ that would be wrong if the true graph were $\mathcal{G}_{\text{true}}$. It allows for direct comparison between a learned causal order and the true DAG. It offers an alternative to existing approaches, such as rank correlations or order distances that lower bound the SHD [Rolland et al., 2022] and that may be difficult to interpret or computationally expensive because in general the causal order of a graph is neither unique nor a total order.

## 6   IMPLEMENTATION

We sketch our implementation of the distances for CPDAG inputs with $p$ nodes and $m$ edges; see Appendix D for details. First, consider a single tuple $(T, Y)$. For the identification strategy, we check whether $(\mathcal{G}_{\text{guess}}, T, Y)$ is amenable and if so compute a) $\text{Pa}(T, \mathcal{G}_{\text{guess}})$ for the Parent-AID, b) $\text{De}(T, \mathcal{G}_{\text{guess}})$ and $\text{An}(T, \mathcal{G}_{\text{guess}})$ for the Ancestor-AID, or c) $\text{De}(T, \mathcal{G}_{\text{guess}})$ and $\mathbf{O}(T, Y, \mathcal{G}_{\text{guess}})$[5] for the Oset-AID. For the verifier, we check whether a) $(\mathcal{G}_{\text{true}}, T, Y)$ is amenable, b) $Y \in \text{NonDe}(T, \mathcal{G}_{\text{true}})$, or c) the proposed adjustment set is a valid adjustment set for $(T, Y)$ in $\mathcal{G}_{\text{true}}$. Algorithms exist to perform each of these steps in $O(p + m)$ time [van der Zander et al., 2014], so we can compute the distances in $O(p^2(p + m))$ time by iterating over all tuples.

We improve this complexity for the Parent- and Ancestor-AID by sharing computations between tuples instead of evaluating identification strategy and verifier for each of the $p(p - 1)$ tuples separately. For this we use reachability algorithms, which are inspired by the Bayes-Ball algorithm [Geiger et al., 1989, Shachter, 1998]: They start from a node set, walk along edges per fixed rules, and return the set of all reached nodes [cf. Wienöbst et al., 2024, Appendix C]. Reachability algorithms find all nodes with a certain property that depends on the rules used. For example, given $T$ and the rule to continue only along $\rightarrow$ edges, the search algorithm finds all nodes in $\text{De}(T, \mathcal{G})$ in $O(p + m)$ time. There are reachability algorithms to compute $\text{De}(T, \mathcal{G})$, $\text{An}(T, \mathcal{G})$,

---

distances when evaluating the performance of PCALG [Wahl and Runge, 2024].

[5]In Lemma 21, Appendix C, we prove a characterization of the Oset that, given amenability, simplifies its computation.

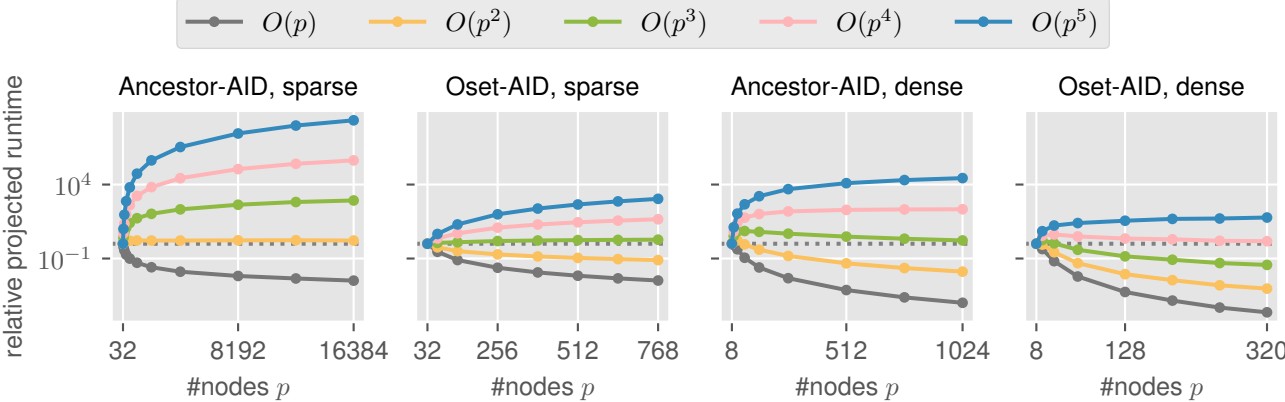

Figure 3: Empirical results on the algorithmic time complexity of calculating the Ancestor-AID $d^{\mathcal{I}_A}$ and the Oset-AID $d^{\mathcal{I}_O}$ between random sparse and dense graphs. We project the runtime under the different time complexities based on the smallest graphs in each panel and visualize the projected runtime as a fraction of the observed empirical runtime; if the relative projected runtime increases/decreases with increasing number of nodes, the considered time complexity suggests a faster/slower increase of runtime than empirically observed. The empirical analysis suggests that our implementation of the Ancestor-AID achieves the time complexity of $O(p^2)$ for sparse and $O(p^3)$ for dense graphs, and that the implementation of the Oset-AID achieves the time complexity of $O(p^3)$ for sparse and $O(p^4)$ for dense graphs. See Appendix E for details.

and similar sets. We develop new walk-status-aware reachability algorithms that, given a graph $\mathcal{G}$, treatment $T$, and candidate adjustment set $\mathbf{Z}$, return a) all nodes such that $(\mathcal{G}, T, Y)$ is amenable, or b) all nodes $Y$ such that $\mathbf{Z}$ is a valid adjustment set for $(T, Y)$ in $\mathcal{G}$ (Algorithms 2 and 3).

These reachability algorithms enable our computationally efficient implementation. When using local adjustment strategies, we can fix a $T$ and compute both the identification strategy and verifier for all $Y$ via at most six reachability algorithms. Selecting each node as $T$ once, we can calculate the Parent- and Ancestor-AID in $O(p(p + m))$ time; this amounts to $O(p^2)$, the optimum, for sparse graphs with $m \in O(p)$ and to $O(p^3)$ for dense graphs. The asymptotic runtime complexity of the Oset-AID remains $O(p^2(p + m))$ since $\mathbf{O}(T, Y, \mathcal{G})$ depends on both $T$ and $Y$.

The original SID implementation has $O(p^4 \log(p))$ runtime for DAGs and exponential runtime for CPDAGs [Peters and Bühlmann, 2015]. Our implementation of the related Parent-AID between either DAGs or CPDAGs has runtime $O(p^3)$ for dense and $O(p^2)$ for sparse graphs. To our knowledge, our distances are the first causal distances between CPDAGs with a polynomial runtime guarantee.

# 7 EMPIRICAL RESULTS

We provide a simulation study quantifying the empirical runtime of our algorithms. In an additional simulation study, we compare the three distances we propose in this paper across various pairs of graphs.

## 7.1 EMPIRICAL RUNTIME ANALYSIS

We calculate distances with our `gadjid` package version 0.1.0, implemented in Rust and using a graph memory layout purposefully designed for fast memory access in reachability algorithms. We use the CRAN SID package v1.1 and run all experiments on a laptop with 8 GB RAM and 4-core i5-8365U processor. We draw DAGs with $p$ nodes, uniformly random total order of nodes, and edges compatible with this order independently drawn with probability $20/(p-1)$ for sparse graphs with $10p$ edges in expectation and 0.3 for dense graphs with $0.3p(p-1)/2$ edges in expectation.

To empirically validate the theoretical asymptotic runtime complexities, we evaluate the Ancestor-AID and the Oset-AID on random DAGs. For each graph size, we record the runtime averaged over 5 repetitions. Based on the runtimes for the smallest graphs, we project what runtimes we would expect for larger graphs under various time complexities. Figure 3 shows the results and Appendix E provides details.

Next, we draw 11 pairs of random DAGs, calculate a distance, and if the median runtime is less than 60 seconds, we increase the number of nodes by one and repeat; we repeat until the median runtime exceeds 60 seconds and obtain:

| Maximum graph size feasible within 1 minute | | |
|---|---|---|
| Method | sparse | dense |
| Parent-AID | 13601 | 962 |
| Ancestor-AID | 8211 | 932 |
| Oset-AID | 1105 | 508 |
| SID | 256 | 239 |

Finally, we consider the graph sizes for which the average runtime of the SID first exceeded one minute, and the extremely sparse graphs from Peters and Bühlmann [2015]; for 11 random pairs of graphs of that size and sparsity, we obtain the following average runtimes:

Average runtime

| Method | x-sparse[6]
$p = 1000$ | sparse
$p = 256$ | dense
$p = 239$ |
|---|---|---|---|
| Parent-AID | 7.3 ms | 30.5 ms | 173 ms |
| Ancestor-AID | 3.4 ms | 40.9 ms | 207 ms |
| Oset-AID | 5.0 ms | 567 ms | 1.68 s |
| SID | ~1–2 h | ~60 s | ~60 s |

## 7.2 DISTANCE COMPARISON

To compare the distances and empirically demonstrate that they capture distinct information, we draw random pairs of graphs and compute the Parent-AID, Ancestor-AID, Oset-AID, and SHD between these pairs. For the distances between 300 pairs of 30-node graphs where $\mathcal{G}_{\text{true}}$ is a random dense graph (sampled as described above) and $\mathcal{G}_{\text{guess}}$ is the graph obtained by removing one edge from $\mathcal{G}_{\text{true}}$ at random, we obtain the following correlation matrix.

| | Ancestor-AID | Oset-AID | Parent-AID |
|---|---|---|---|
| Ancestor-AID | 1 | 0.7281 | 0.0886 |
| Oset-AID | 0.7281 | 1 | 0.2080 |
| Parent-AID | 0.0886 | 0.2080 | 1 |

Further, the average distances are Ancestor-AID: 2.0, Oset-AID: 5.9, and Parent-AID: 11.2 (while the SHD between all these graph pairs is 1). We provide a corresponding scatter plot between the distances in Figure 5, Appendix F.1. The results highlight that the number of wrongly inferred causal effects if we delete an edge from the true DAG, depends on the choice of identification strategy.

When benchmarking causal discovery algorithms, the distance should be chosen in line with the downstream task the graph will eventually be used for. If the task is to reason about the existence of direct cause-effect relationships, the SHD is a natural choice. If the task is to infer causal effects, there are multiple options. The Parent-, Ancestor-, and Oset-AID are three such options, each corresponding to different assumptions on the behavior of an idealized practitioner who will use $\mathcal{G}_{\text{guess}}$ to infer causal effects. This simulation

---

[6]We denote the sparse graphs with $0.75p$ expected edges considered in Peters and Bühlmann [2015] as extremely sparse (x-sparse); for x-sparse 1000-node random graphs, Peters and Bühlmann [2015] reported a runtime of almost 7000 s, which on our hardware took ~1 h (running 1 instead of 11 repetitions).

experiment and additional experiments in Appendix F underline that the choice of distance is practically important when benchmarking causal discovery algorithms.

## 8 DISCUSSION

Our framework gives a recipe for developing distances for other graph types, such as maximal ancestral graphs that allow for hidden variables: Find a sound and complete identification strategy and a corresponding verifier. While the adjustment-based identification strategies we use for DAGs and CPDAGs are not sound and complete for settings with hidden variables, sound and complete alternatives exist [Huang and Valtorta, 2006, Shpitser and Pearl, 2006]. Yet, there are no verifiers for these alternatives. Therefore, advances on causal effect identification and in particular verification are needed before we can develop distances for other graph types so as to aid the development of causal discovery under latent confounding. Nonetheless, the framework for strategy-specific identification distances provides a handbook on how to develop such a distance as the necessary methodology for causal effect identification and verification becomes available.

### Acknowledgements

We thank Alexander G. Reisach for valuable discussions and feedback on an earlier draft of the present manuscript. We also thank the anonymous reviewers for constructive feedback that helped improve the presentation.

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

# Adjustment Identification Distance: A `gadjid` for Causal Structure Learning (Supplementary Material)

**Leonard Henckel**[1]  **Theo Würtzen**[2]  **Sebastian Weichwald**[2,3]

[1]School of Mathematics and Statistics, University College Dublin, Ireland
[2]Pioneer Centre for AI, University of Copenhagen, Denmark
[3]Department of Mathematical Sciences, University of Copenhagen, Denmark

## A  ADDITIONAL PRELIMINARIES

**Simple graphs with directed and undirected edges.**  A simple graph $\mathcal{G} = (\mathbf{V}, \mathbf{E})$ over nodes $\mathbf{V} = \{V_i \mid i \in [d]\}$ with edges $\mathbf{E}$ is a graph where there is at most one edge between any two nodes. A graph is directed, if all edges are directed edges $\rightarrow$, and partially directed, if all edges are directed edges $\rightarrow$ or undirected edges $-$. Two nodes are adjacent, if an edge connects them. In particular a node is adjacent to itself.

**Walks and paths.**  A walk $w$ is a sequence of nodes $(T, ..., Y)$ where each successive pair of nodes is adjacent. The nodes $T$ and $Y$ are called endpoint nodes on $w$. A path $p$ is a sequence of distinct nodes $(T, ..., Y)$ and is a special case of a walk. A walk $w$ is possibly directed from $T$ to $Y$ if no directed edge along the path is directed towards $T$ (a possibly directed walk is sometimes called possibly causal). A walk $w$ is directed from $T$ to $Y$ if all edges along the path are directed and directed towards $Y$ (a directed walk is sometimes called causal); every directed walk is also a possibly directed walk. We often consider walks that are not possibly directed as they contain at least one edge facing towards $T$, for ease, we call such a walk $w$ non-causal. A directed path from $T$ to $Y$ together with $Y \rightarrow T$ forms a cycle.

**DAGs and PDAGs.**  A directed acyclic graph (DAG) is a simple directed graph, that is, all edges are directed, that has no cycles. A partially directed acyclic graph (PDAG) is a simple graph that is partially directed and has no cycles. A DAG is also a PDAG. A walk $w$ from a set $\mathbf{T}$ to a set $\mathbf{Y}$ is a walk from some node $T \in \mathbf{T}$ to some node $Y \in \mathbf{Y}$, that is, $T$ and $Y$ are the endpoint nodes of the walk. The walk $w$ is called proper, if it only contains one node in $\mathbf{T}$. Given two walks $w = (A, \ldots, B)$ and $w' = (B, \ldots, C)$ we let $w \oplus w' = (A, \ldots, B, \ldots, C)$ denote the walk we obtain by concatenating $w$ and $w'$.

**Node relationships in DAGs and PDAGs.**  If the edge $T \rightarrow Y$ or $T - Y$ exists, $T$ is a possible parent of $Y$ and $Y$ a possible child of $T$. Let $\mathrm{PossPa}(Y, \mathcal{G})$ denote the set of all possible parents of $Y$ and $\mathrm{PossCh}(T, \mathcal{G})$ the set of all possible children of $T$. If there is a possibly directed path from $T$ to $Y$ or if $T = Y$, $T$ is a possible ancestor of $Y$ and $Y$ a possible descendant of $T$. Let $\mathrm{PossAn}(Y, \mathcal{G})$ denote the set of all possible ancestors of $Y$ and $\mathrm{PossDe}(T, \mathcal{G})$ the set of all possible descendants of $T$. If all edges are directed, we analogously define the set of parents $\mathrm{Pa}(Y, \mathcal{G})$, children $\mathrm{Ch}(T, \mathcal{G})$, ancestors $\mathrm{An}(Y, \mathcal{G})$, and descendants $\mathrm{De}(T, \mathcal{G})$. For a set $\mathbf{T}$ we define $\mathrm{PossPa}(\mathbf{T}, \mathcal{G}) = \bigcup_{T \in \mathbf{T}} \mathrm{PossPa}(T, \mathcal{G})$; we analogously define $\mathrm{PossCh}(\mathbf{T}, \mathcal{G}), \mathrm{PossAn}(\mathbf{T}, \mathcal{G}), \mathrm{PossDe}(\mathbf{T}, \mathcal{G}), \mathrm{Ch}(\mathbf{T}, \mathcal{G}), \mathrm{Pa}(\mathbf{T}, \mathcal{G}), \mathrm{De}(\mathbf{T}, \mathcal{G})$, and $\mathrm{An}(\mathbf{T}, \mathcal{G})$. We also define $\mathrm{NonDe}(T, \mathcal{G}) = \mathbf{V} \setminus \mathrm{PossDe}(T, \mathcal{G})$ which in the DAG case reduces to $\mathrm{NonDe}(T, \mathcal{G}) = \mathbf{V} \setminus \mathrm{De}(T, \mathcal{G})$.

**Supergraph.**  A graph $\mathcal{G} = (\mathbf{V}, \mathbf{E})$ is a called a supergraph of a graph $\mathcal{G}' = (\mathbf{V}, \mathbf{E}')$ if $\mathbf{E}' \subseteq \mathbf{E}$. We say that $\mathcal{G}$ is a super-DAG of $\mathcal{G}'$ if $\mathcal{G}$ is a DAG, and define super-CPDAG analogously.

**Colliders, v-structures, and definite-status paths.**  A node $V$ in a PDAG $\mathcal{G}$ is a collider on a path $p$ if $p$ contains the subpath $U \rightarrow V \leftarrow W$. Node $V$ on a path $p$ is called a definite non-collider on $p$ if $p$ contains a subpath such that (a) $U \leftarrow V$, (b) $V \rightarrow W$, or (c) $U - V - W$ and $U$ and $W$ are not adjacent in $\mathcal{G}$. A path $p$ is of definite status if every node on $p$ is a collider, a definite-status non-collider, or an endpoint node. We define all terms analogously for walks. A path of the

form $U \to V \leftarrow W$ in $\mathcal{G}$ is called a v-structure if $U$ and $V$ are not adjacent in $\mathcal{G}$.

**Blocking and d-separation in PDAGs.** Let $\mathbf{Z}$ be a set of nodes in a PDAG $\mathcal{G}$. A definite-status path $p$ is blocked given $\mathbf{Z}$ if $p$ either contains a non-collider $N \in \mathbf{Z}$ or a collider $C$ such that $\mathrm{De}(C, \mathcal{G}) \cap \mathbf{Z} = \emptyset$. A definite-status walk $w$ is blocked given $\mathbf{Z}$ if $p$ either contains a non-collider $N \in \mathbf{Z}$ or a collider $C$ such that $C \notin \mathbf{Z}$. A definite-status path or walk that is not blocked given $\mathbf{Z}$ is said to be open or d-connecting given $\mathbf{Z}$. Given three pairwise disjoint node sets $\mathbf{T}, \mathbf{Y}, \mathbf{Z}$ in a PDAG $\mathcal{G}$ we say that $\mathbf{T}$ is d-separated from $\mathbf{Y}$ given $\mathbf{Z}$ in $\mathcal{G}$ and write $\mathbf{T} \perp_{\mathcal{G}} \mathbf{Y} \mid \mathbf{Z}$ if $\mathbf{Z}$ blocks all definite-status paths from $\mathbf{T}$ to $\mathbf{Y}$ or equivalently all definite-status walks. If it does not, we say that $\mathbf{T}$ and $\mathbf{Y}$ are d-connected given $\mathbf{Z}$.

**CPDAGs, Markov property, and Markov equivalence.** A density $f$ is Markov with respect to a DAG $\mathcal{G}$ if for any three pairwise disjoint node sets $\mathbf{T}, \mathbf{Y}, \mathbf{Z}$ in $\mathcal{G}$ such that $\mathbf{T} \perp_{\mathcal{G}} \mathbf{Y} \mid \mathbf{Z}$, $\mathbf{T}$ is conditionally independent of $\mathbf{Y}$ given $\mathbf{Z}$. Two DAGs that encode the same set of d-separation statements are called Markov equivalent. Given a DAG $\mathcal{G}$, the set of all DAGs that are Markov equivalent to $\mathcal{G}$ is called a Markov equivalence class and can be represented by a completed partially directed acyclic graph (CPDAG); a special subset of PDAGs characterized by Meek [1995]. Note that a DAG is in general not a CPDAG. Given a CPDAG $\mathcal{C}$ let $[\mathcal{C}]$ denote the corresponding equivalence class. For any DAG $\mathcal{D} \in [\mathcal{C}]$ we say that $\mathcal{C}$ is the CPDAG of $\mathcal{D}$. The CPDAG $\mathcal{C}$ has the same v-structures and adjacencies as any DAG in $[\mathcal{C}]$. Further, an edge $\to$ in $\mathcal{C}$ implies that every DAG in $[\mathcal{C}]$ contains that edge $\to$. An edge — in $\mathcal{C}$ implies that some DAG in $[\mathcal{C}]$ contains the edge $\to$ and some other DAG in $[\mathcal{C}]$ contains the edge $\leftarrow$.

**Causal and forbidden Nodes.** Given set $\mathbf{T}$ and $\mathbf{Y}$ in a DAG or CPDAG $\mathcal{G}$, we define the causal nodes $\mathrm{Cn}(\mathbf{T}, \mathbf{Y}, \mathcal{G})$ to be all nodes that lie on proper directed paths from $\mathbf{T}$ to $\mathbf{Y}$ and are not in $\mathbf{T}$. In a CPDAG $\mathcal{G}$, we define the possibly causal nodes $\mathrm{PossCn}(\mathbf{T}, \mathbf{Y}, \mathcal{G})$ to be all nodes that lie on proper possibly directed paths from $\mathbf{T}$ to $\mathbf{Y}$ and are not in $\mathbf{T}$. We define the forbidden nodes as $\mathrm{Forb}(\mathbf{T}, \mathbf{Y}, \mathcal{G}) = \mathrm{PossDe}(\mathrm{PossCn}(\mathbf{T}, \mathbf{Y}, \mathcal{G}), \mathcal{G}) \cup \mathbf{T}$.

**Amenability.** Let $\mathbf{T}$ and $\mathbf{Y}$ be disjoint node sets in a DAG, CPDAG, MAG or PAG $\mathcal{G}$. We say that $(\mathcal{G}, \mathbf{T}, \mathbf{Y})$ is amenable if every proper possibly directed path from $\mathbf{T}$ to $\mathbf{Y}$ begins with an edge $\to$.

The generalized adjustment criterion by Perkovic et al. [2018] provides necessary and sufficient graphical conditions for a set to be a valid adjustment set:

**Definition 17** (*Generalized Adjustment Criterion*). Let $\mathbf{T}, \mathbf{Y}$, and $\mathbf{Z}$ be pairwise disjoint node sets in a DAG, CPDAG, MAG or PAG $\mathcal{G}$. Then $\mathbf{Z}$ satisfies the generalized adjustment criterion relative to $(\mathbf{T}, \mathbf{Y})$ in $\mathcal{G}$ if the following three conditions hold:

**(Amenability)** $(\mathcal{G}, \mathbf{T}, \mathbf{Y})$ is amenable, and

**(Forbidden set)** $\mathbf{Z} \cap \mathrm{Forb}(\mathbf{T}, \mathbf{Y}, \mathcal{G}) = \emptyset$, and

**(Blocking)** all proper definite-status non-causal paths from $\mathbf{T}$ to $\mathbf{Y}$ are blocked by $\mathbf{Z}$ in $\mathcal{G}$.

# B  PROOFS

*Proof of Proposition 7 (Distance to Super-DAG is Zero).* Let $I = \mathcal{I}(\mathcal{G}_{\mathrm{guess}}, T, Y)$ be an identifying formula. Since $\mathcal{I}$ is sound and complete, $I = f(\mathbf{y} \mid \mathrm{do}(\mathbf{t}))$ for any $f$ compatible with $\mathcal{G}_{\mathrm{guess}}$. Since any $f$ compatible with $\mathcal{G}_{\mathrm{guess}}$ is also compatible with $\mathcal{G}_{\mathrm{true}}$ (since all parent sets in $\mathcal{G}_{\mathrm{guess}}$ are supersets of parent sets in $\mathcal{G}_{\mathrm{true}}$) it follows that $I = f(\mathbf{y} \mid \mathrm{do}(\mathbf{t}))$ for any $f$ compatible with $\mathcal{G}_{\mathrm{true}}$. Therefore, $\mathcal{V}(\mathcal{G}_{\mathrm{true}}, \mathcal{I}(\mathcal{G}_{\mathrm{guess}}, T, Y))$ returns correct. Since this is true for any tuple $(T, Y)$ our claim follows. $\square$

*Proof of Lemma 9 (Ancestors are Valid Adjustment Sets).* Since $\mathrm{Forb}(T, Y, \mathcal{G}) \setminus \mathrm{De}(T, \mathcal{G}) = \emptyset$, $\mathbf{Z}$ satisfies the forbidden set condition of Definition 17. Let $p$ be a non-causal path from $T$ to $Y$. If $p$ begins with an edge into $T$, it contains a node in $\mathrm{Pa}(T, \mathcal{G})$ which is therefore also a node in $\mathbf{Z}$ that is a non-collider on $p$. It follows that $p$ is blocked. If $p$ begins with an edge out of $T$, $p$ is either directed or it must contain a collider $C \in \mathrm{De}(T, \mathcal{G})$. Since $\mathbf{Z} \cap \mathrm{De}(T, \mathcal{G}) = \emptyset$ and $\mathrm{De}(C, \mathcal{G}) \subseteq \mathrm{De}(T, \mathcal{G})$ it follows that $p$ is blocked given $\mathbf{Z}$. Therefore, $\mathbf{Z}$ satisfies the adjustment criterion. $\square$

*Proof of Lemma 10 (Parent-AID Misrepresents Causal Order).* Here, $\mathrm{Pa}(V_t, \mathcal{G}_{\mathrm{guess}}) = V_{t-1}$ for $t - 1 \geq 1$ and therefore

the parent adjustment strategy $\mathcal{I}_P(\mathcal{G}_{\text{guess}}, V_t, V_y)$ returns

$$\begin{cases} \int f(v_y \mid v_t, v_{t-1}) f(v_{t-1}) \, \mathrm{d}v_{t-1} & \text{if } y \neq t-1 \geq 1, \\ f(v_y \mid v_t) \ (\text{``empty adjustment set''}) & \text{if } \quad t-1 = 0, \\ f(v_y) & \text{if } y = t-1 \geq 1. \end{cases}$$

We now apply the verifier $\mathcal{V}^{\mathcal{D}}_{\text{adj}}$. Consider first inputs to the verifier of the form $I = f(v_y)$. The DAGs $\mathcal{G}_{\text{true}}$ and $\mathcal{G}_{\text{guess}}$ have the same causal ordering and therefore any one of the $p-1$ such inputs with $y = t-1 \geq 1$ is correct. Consider now the remaining inputs to the verifier of the adjustment form. Since $V_i \in \text{Pa}(V_t, \mathcal{G}_{\text{true}})$ for all $i < t-1$, no valid adjustment set exists for the effect of $V_t$ on such a $V_i$ in $\mathcal{G}_{\text{true}}$. Further, if $t < y$ then the only valid adjustment set is $\{V_1, \ldots, V_{t-1}\}$. Therefore, $\text{Pa}(V_t, \mathcal{G}_{\text{guess}})$ is a valid adjustment set in $\mathcal{G}_{\text{true}}$ if and only if $t = 1$ or $t = 2$ and $y > t$. Using $\mathcal{I}_P$ we therefore obtain exactly $3p-4 = (p-1) + (p-1) + (p-2)$ identification formulas in $\mathcal{G}_{\text{guess}}$ that are also correct in $\mathcal{G}_{\text{true}}$, while the remaining $(p^2 - p) - (3p-4) = p^2 - 4p + 4$ are false. It follows that

$$\lim_{p \to \infty} d^{\mathcal{I}_P}(\mathcal{G}_{\text{guess}}, \mathcal{G}_{\text{true}}) / (p^2 - p) = 1.$$

$\square$

*Proof of Proposition 12 (Ancestor-AID Reflects Causal Order).* Consider two nodes $T$ and $Y$ in a DAG $\mathcal{G}_{\text{guess}}$. If $Y \in \mathbf{D}_T = \text{De}(T, \mathcal{G}_{\text{guess}})$, then $\mathbf{A}_T = \text{An}(T, \mathcal{G}_{\text{guess}})$ is a valid adjustment set in $\mathcal{G}_{\text{guess}}$ by Lemma 9. If $Y \notin \mathbf{D}_T$, then $f(y \mid \text{do}(t)) = f(y)$. Therefore, $\mathcal{I}_A$ is sound and complete.

Consider two DAGs $\mathcal{G}_{\text{true}}$ and $\mathcal{G}_{\text{guess}}$, such that $\mathcal{G}_{\text{guess}}$ respects the causal orders of $\mathcal{G}_{\text{true}}$, that is, $\text{De}(T, \mathcal{G}_{\text{true}}) \subseteq \text{De}(T, \mathcal{G}_{\text{guess}})$ or equivalently $\text{An}(T, \mathcal{G}_{\text{true}}) \subseteq \text{An}(T, \mathcal{G}_{\text{guess}})$ for all nodes $T$. Fix a pair $(T, Y)$ and consider $I_{TY} = \mathcal{I}_A(\mathcal{G}, T, Y)$. If $Y \notin \text{De}(T, \mathcal{G}_{\text{guess}})$ then $Y \notin \text{De}(T, \mathcal{G}_{\text{true}})$ and therefore, $I_{TY} = f(y)$ is a correct identifying formula in $\mathcal{G}_{\text{true}}$. If $Y \in \text{De}(T, \mathcal{G}_{\text{guess}})$ then $Y \notin \text{An}(T, \mathcal{G}_{\text{guess}})$, which implies that $Y \notin \text{An}(T, \mathcal{G}_{\text{true}})$ and $Y \notin \text{Pa}(T, \mathcal{G}_{\text{true}})$. Since $\text{Pa}(T, \mathcal{G}_{\text{true}}) \subseteq \text{An}(T, \mathcal{G}_{\text{true}}) \subseteq \text{An}(T, \mathcal{G}_{\text{guess}}) \subseteq \text{NonDe}(T, \mathcal{G}_{\text{guess}})$ and $\text{NonDe}(T, \mathcal{G}_{\text{guess}}) = \mathbf{V} \setminus \text{De}(T, \mathcal{G}_{\text{guess}}) \subseteq \mathbf{V} \setminus \text{De}(T, \mathcal{G}_{\text{true}})$ we can therefore invoke Lemma 9 to conclude that $I_{TY}$ is correct in $\mathcal{G}_{\text{true}}$. Therefore, $d^{\mathcal{I}_A}(\mathcal{G}_{\text{true}}, \mathcal{G}_{\text{guess}}) = 0$.

Suppose now that $\mathcal{G}_{\text{guess}}$ does not respect the causal order of $\mathcal{G}_{\text{true}}$, that is, there exists a pair $T, Y$ such that $Y \in \text{De}(T, \mathcal{G}_{\text{true}}) \setminus \text{De}(T, \mathcal{G}_{\text{guess}})$. Since $Y \notin \text{De}(T, \mathcal{G}_{\text{guess}})$, $I_{TY} = f(y)$ but this is wrong in $\mathcal{G}_{\text{true}}$. $\square$

*Proof of Proposition 15 (Oset-AID is the $\mathcal{I}_O$-Specific Distance).* The soundness and completeness of $\mathcal{I}_O$ follows by the results of Henckel et al. [2022] and that the causal effect on a non-descendant is the observational density. $\square$

# C  ADDITIONAL EXAMPLES AND RESULTS

**Example 18** (*Counter-Examples where Oset Adjustment Distance is (not) Zero*). Let $\mathcal{G}^1_{\text{true}}$ be the graph from Figure 4a, $\mathcal{G}^2_{\text{true}}$ be the graph from Figure 4c, and $\mathcal{G}_{\text{guess}}$ be the graph from Figure 4b.

Consider the pair $(\mathcal{G}^1_{\text{true}}, \mathcal{G}_{\text{guess}})$. Since, $V_2$ is an isolated node in $\mathcal{G}^1_{\text{true}}$ any identifying formula produced by $\mathcal{I}_O$ for a pair involving $V_2$ will be correct irrespective of $\mathcal{G}_{\text{guess}}$. Further, $\mathcal{I}_O(\mathcal{G}_{\text{guess}}, V_3, V_1) = f(v_1)$ and $\mathcal{I}_O(\mathcal{G}_{\text{guess}}, V_1, V_3) = f(v_3 \mid v_1)$. Since $V_1 \notin \text{De}(V_3, \mathcal{G}^1_{\text{true}})$ and the empty set is a valid adjustment set relative to $(V_1, V_3)$ in $\mathcal{G}^1_{\text{true}}$ it follows that $d^{\mathcal{I}_O}(\mathcal{G}^1_{\text{true}}, \mathcal{G}_{\text{guess}}) = 0$. This shows that the Oset adjustment distance may be zero even if $\mathcal{G}_{\text{guess}}$ is not a supergraph of $\mathcal{G}_{\text{true}}$.

Consider the pair $(\mathcal{G}^2_{\text{true}}, \mathcal{G}_{\text{guess}})$. Since $I_O(\mathcal{G}_{\text{guess}}, V_2, V_3) = f(v_3 \mid v_2)$ and the empty set is not a valid adjustment set relative to $(V_2, V_3)$ in $\mathcal{G}^2_{\text{true}}$, $d^{\mathcal{I}_O}(\mathcal{G}^2_{\text{true}}, \mathcal{G}_{\text{guess}}) \neq 0$. This shows that $\mathcal{G}_{\text{guess}}$ respecting the causal order of $\mathcal{G}_{\text{true}}$ does not ensure that the Oset adjustment distance is zero.

**Example 19** (*Correct CPDAG may be further from true DAG than Incorrect CPDAG*). Let $\mathcal{G}^D_{\text{true}}$ be a fully connected DAG. Let $\mathcal{G}^1_{\text{guess}}$ be the corresponding CPDAG, that is, the fully connected CPDAG. Since every effect in $\mathcal{G}_{\text{guess}}$ is non-identifiable it follows that for any strategy-specific distance $d^{\mathcal{I}}(\mathcal{G}_{\text{true}}, \mathcal{G}^1_{\text{guess}}) = p(p-1)$. Let $\mathcal{G}^2_{\text{guess}}$ be the empty CPDAG. Since exactly half the effects in $\mathcal{G}_{\text{true}}$ are zero it follows that for any identification strategy that uses a descendant check, such as $\mathcal{I}_A$ or $\mathcal{I}_O$, $d^{\mathcal{I}}(\mathcal{G}_{\text{true}}, \mathcal{G}^2_{\text{guess}}) = p(p-1)/2$

**Proposition 20** (Ancestor Adjustment Strategy is Sound and Complete for CPDAGs). *Consider nodes $T$ and $Y$ in a CPDAG $\mathcal{G}$, such that $(\mathcal{G}, T, Y)$ is amenable. Then $\text{An}(T, \mathcal{G})$ is a valid adjustment set relative to $(T, Y)$ in $\mathcal{G}$.*

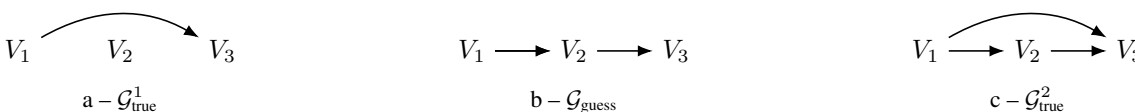

$\text{a} - \mathcal{G}^1_{\text{true}}$      $\text{b} - \mathcal{G}_{\text{guess}}$      $\text{c} - \mathcal{G}^2_{\text{true}}$

Figure 4: DAGs for Example 18.

*Proof.* Since by assumption $(\mathcal{G}, T, Y)$ is amenable and $\text{An}(T, \mathcal{G})$ satisfies the forbidden set condition of Definition 17, it only remains to show that $\text{An}(T, \mathcal{G})$ satisfies the blocking condition. To see this consider a definite-status non-causal path $p$ from $T$ to $Y$. If $p$ begins with an edge $\leftarrow$ it contains a node in $\text{Pa}(T, \mathcal{G})$ and is therefore blocked given $\text{An}(T, \mathcal{G})$. If it begins with an edge $-$ or $\rightarrow$, then it must contain at least one collider $C \in \text{PossDe}(T, \mathcal{G})$ by the fact that it is of definite status and may therefore not contain $-V \leftarrow$ but contains at least one backwards facing edge. Since $\text{De}(C, \mathcal{G}) \cap \text{An}(T, \mathcal{G}) = \emptyset$, it follows that $p$ is blocked given $\text{An}(T, \mathcal{G})$. $\qquad\square$

**Lemma 21** (Oset-Characterization Simplifies Given Amenability). *Consider two node sets $\mathbf{T}$ and $\mathbf{Y}$ in a CPDAG $\mathcal{G}$ such that $(\mathcal{G}, \mathbf{T}, \mathbf{Y})$ is amenable. Then*

$$\mathbf{O}(\mathbf{T}, \mathbf{Y}, \mathcal{G}) = \text{Pa}(\text{De}(\mathbf{T}, \mathcal{G}) \cap \text{PropAn}(\mathbf{Y}, \mathbf{T}, \mathcal{G}), \mathcal{G}) \setminus \text{De}(\mathbf{T}, \mathcal{G}),$$

*where $\text{PropAn}(\mathbf{Y}, \mathbf{T}, \mathcal{G})$ is the set of all nodes $N$, such that there exists a directed path from $N$ to some $Y \in \mathbf{Y}$ that does not contain any nodes in $\mathbf{T}$.*

*Proof.* Recall that $\mathbf{O}(\mathbf{T}, \mathbf{Y}, \mathcal{G}) = \text{Pa}(\text{Cn}(\mathbf{T}, \mathbf{Y}, \mathcal{G}), \mathcal{G}) \setminus \text{Forb}(\mathbf{T}, \mathbf{Y}, \mathcal{G})$. Clearly,

$$\text{Cn}(\mathbf{T}, \mathbf{Y}, \mathcal{G}) = \text{De}(\mathbf{T}, \mathcal{G}) \cap \text{PropAn}(\mathbf{Y}, \mathbf{T}, \mathcal{G})$$

by the definition of $\text{Cn}(\mathbf{T}, \mathbf{Y}, \mathcal{G})$. Since every causal node is in $\text{PropAn}(\mathbf{Y}, \mathbf{T}, \mathcal{G})$ so is every node in $\mathbf{O}(\mathbf{T}, \mathbf{Y}, \mathcal{G})$. Since every node that is both in $\text{PropAn}(\mathbf{Y}, \mathbf{T}, \mathcal{G})$ and $\text{De}(\mathbf{T}, \mathcal{G})$ is a causal node and therefore forbidden it follows that $\mathbf{O}(\mathbf{T}, \mathbf{Y}, \mathcal{G}) \cap \text{De}(\mathbf{T}, \mathcal{G}) = \emptyset$. By Lemma E.6 by Henckel et al. [2022], $\text{Forb}(\mathbf{T}, \mathbf{Y}, \mathcal{G}) \subseteq \text{De}(\mathbf{T}, \mathcal{G})$ and therefore removing all nodes in $\text{De}(\mathbf{T}, \mathcal{G})$ from $\text{Pa}(\text{Cn}(\mathbf{T}, \mathbf{Y}, \mathcal{G}), \mathcal{G})$ is equivalent to removing all nodes in $\text{Forb}(\mathbf{T}, \mathbf{Y}, \mathcal{G})$. $\qquad\square$

# D    ALGORITHM DEVELOPMENT

Many graph properties involved in the validation of adjustment sets, such as amenability, forbidden nodes, or blocking, are based on the (non-)existence of paths with certain properties. For example, two nodes $T$ and $Y$ are d-connected in a DAG given $\mathbf{Z}$ if and only if a path exists between them that is not blocked by $\mathbf{Z}$. We can reformulate the problem of verifying whether such a path exists as a reachability task: Starting from $T$, try reaching $Y$ by following all possible paths that are not blocked by $\mathbf{Z}$ until either you reach $Y$ or you have exhausted all possible paths.

The *Bayes-Ball* algorithm [Geiger et al., 1989, Shachter, 1998] uses the reachability framework to obtain all nodes in a DAG that are d-connected to $T$ given $\mathbf{Z}$. A key insight for its efficient implementation is that a d-connecting walk exists between two nodes if and only if a d-connecting path exists between them; thus, we can avoid having to a) check for each collider along the path that one of its descendants is in $\mathbf{Z}$ before continuing and to b) follow all possible paths for which we need to keep track of all previously visited nodes along any given path traversal (to avoid visiting the same node more than once). Instead, we can traverse along walks and continue a walk from $V$ to $W$ along an incoming/outgoing edge if a') $W$ has not previously been reached through an incoming/outgoing edge and if b') the walk is not blocked by $\mathbf{Z}$ (if we face $\rightarrow V \leftarrow W$, we continue from $V$ to $W$ only if $V \in \mathbf{Z}$, otherwise we continue only if $V \notin \mathbf{Z}$). The benefit of a') and b') over a) and b) is that both conditions are local to the current node in a walk and verifiable without querying ancestor sets or storing and checking against all previously visited nodes. In this walk reachability algorithm, each node is visited at most twice and each edge considered a constant number of times [see, for example, Appendix A in Wienöbst et al., 2024] and therefore its runtime is linear in the number of nodes $p$ plus the number of edges $m$; in dense graphs the number of edges grows quadratic in the number of nodes and consequently the runtime is $O(p^2)$ for dense graphs.

Wienöbst et al. [2024] generalize the algorithmic concept underlying *Bayes-Ball* to a class of DAG search algorithms akin to a depth-first graph search that recursively visits neighbouring nodes that have not been reached by the same kind of edge before. Since each node $V$ is visited at most once per edge type (for example, $\rightarrow V$ or $\leftarrow V$ in DAGs), the runtime of their

*gensearch* algorithm is also $O(p^2)$. Rule tables encode the conditions for continuing on a given walk based on how the current node was reached and how the potential next node $W$ would be reached. Which rule table we use for the *gensearch* algorithm determines the properties of the nodes that will be reached and therefore returned by the algorithm; for example, they show that a sequence of *gensearch* algorithms with carefully chosen rule tables finds a minimal adjustment set in $O(p + m)$ time.

Key to implementing our adjustment distances efficiently, is a new walk-status-aware reachability algorithm that, given a DAG or CPDAG $\mathcal{G}$ with $p$ nodes and $m$ edges, set $\mathbf{Z}$, and treatment nodes $\mathbf{T}$, returns all nodes $Y$ such that $\mathbf{Z}$ is a valid adjustment set for $(\mathbf{T}, Y)$ in $\mathcal{G}$ in $O(p + m)$ time. We use this algorithm to verify an adjustment set for many $Y$ simultaneously. To implement this algorithm, we prove a modified adjustment criterion, adapt the reachability algorithm for finding d-connected nodes in DAGs to account for walks that are not of definite-status in CPDAGs, implement an amenability check, and show how to find nodes that satisfy all conditions of the modified adjustment criterion with only one reachability algorithm. We proceed as follows:

- In Appendix D.1 we prove a modified adjustment criterion that translates all conditions of the generalized adjustment criterion [Perkovic et al., 2018] into conditions on the (non-)existence of certain walks. An adjustment criterion in terms of walks allows us to verify whether it holds using only reachability algorithms.

- In Appendix D.2 we show how to verify blocking in CPDAGs with a reachability algorithm that uses no non-local information to verify whether a walk is definite-status or not.

- In Appendix D.3 we provide motivation and intuition for our decision to add a walk-status to reachability algorithms that is propagated forward in the depth-first search traversal of the graph. The addition of a walk status allows us to track walks that do not transmit reachability, but may change status and become walks for which we need to track nodes reached by such a walk.

- In Appendix D.4, we demonstrate how we use our new reachability algorithm to calculate the Parent- and Ancestor-AID between DAGs or CPDAGs with $O(p(p + m))$ and the Oset-AID with $O(p^2(p + m))$ time complexity.

## D.1    MODIFIED ADJUSTMENT CRITERION

We prove a modified adjustment criterion that translates all conditions of the generalized adjustment criterion [Perkovic et al., 2018] into conditions on the (non-)existence of certain walks. Having an adjustment criterion exclusively in terms of walks allows us to use reachability algorithms to verify it.

**Lemma 22** (Modified Adjustment Criterion for Walk-Based Verification). *Consider nodes $\mathbf{T}$ and $\mathbf{Y}$ in a DAG or CPDAG $\mathcal{G}$ and a node set $\mathbf{Z}$ in $\mathcal{G}$. The set $\mathbf{Z}$ fulfills the adjustment criterion if and only if*

1. *every proper possibly directed walk from $\mathbf{T}$ to $\mathbf{Y}$ begins with a directed edge out of $\mathbf{T}$, and*

2. *no proper possibly directed walk from $\mathbf{T}$ to $\mathbf{Y}$ contains a node in $\mathbf{Z}$, and*

3. *every proper definite-status walk from $\mathbf{T}$ to $\mathbf{Y}$ that contains a backwards facing edge is blocked by $\mathbf{Z}$.*

*Proof.* We prove our claims for the CPDAG case as the DAG case can be shown with the same basic arguments but is simpler. We first show that if $\mathbf{Z}$ does not satisfy the adjustment criterion then it also does not satisfy the alternative adjustment criterion. Since the two criteria both assume amenability we can assume amenability holds. Suppose that there exists a proper definite-status non-causal path $p$ from $\mathbf{T}$ to $\mathbf{Y}$ that is open given $\mathbf{Z}$. Consider all colliders $C_1, \ldots, C_k$ and the corresponding directed paths $q_1, \ldots, q_k$ from $C_i$ to some node $Z_i \in \mathbf{Z}$. If any of the $q_i$ contains a node in $T' \in \mathbf{T}$ we can replace $p$ with $q_i(T', C_i) \oplus p(C_i, Y)$ so without loss of generality we can assume this is not the case. By appending the $q_i$ to $p$ we obtain a proper definite-status walk from $\mathbf{T}$ to $\mathbf{Y}$ that contains an edge $\leftarrow$, inherited from $p$.

We can therefore assume, no such $p$ exists and that $\mathbf{Z} \cap \text{Forb}(\mathbf{T}, \mathbf{Y}, \mathcal{G}) \neq \emptyset$. Since $\text{Forb}(\mathbf{T}, \mathbf{Y}, \mathcal{G}) = \text{PossDe}(\text{PossCn}(\mathbf{T}, \mathbf{Y}, \mathcal{G}), \mathcal{G})$ we can in fact assume that there exists a node $Z \in \mathbf{Z} \cap (\text{PossDe}(\text{PossCn}(\mathbf{T}, \mathbf{Y}, \mathcal{G}), \mathcal{G}) \setminus \text{PossCn}(\mathbf{T}, \mathbf{Y}, \mathcal{G}))$. For such a node $Z$ there exists a directed path $p_1$ from $T$ to $Z$ by Lemma E.6 of Henckel et al. [2022] and a possibly directed path $p_2$ from some causal node $N$. Since $N$ is possibly causal there also exists a possibly directed path $p_3$ from $N$ to some node in $Y \in \mathbf{Y}$. We can choose all three paths to not contain other nodes in $\mathbf{Z}$. Let $I$ be the node closest to $Z$, where $p_2$ and $p_3$ intersect and consider $p_4 = p_2(Z, I) \oplus p_3(I, Y)$. Note that $p_4$ is colliderless and by taking shortcuts we obtain a definite-status path $p_4^*$, that is also colliderless. By assumption on $Z$, $p_4^*$ must be non-causal and therefore the walk $w = p_1 \oplus p_4^*$ has $Z$ as a definite-status collider, contains no other node in $\mathbf{Z}$ and all other nodes

are endpoint nodes or definite-status non-colliders. The walk $w$ therefore violates the blocking condition of the modified criterion.

We now show that if $\mathbf{Z}$ satisfies the adjustment criterion then it satisfies the alternative adjustment criterion. Again we can assume amenability holds. It suffices to show that any proper non-causal definite-status walk $w$ from $T \in \mathbf{T}$ to $Y \in \mathbf{Y}$ is blocked given $\mathbf{Z}$. Suppose $w$ is colliderless. This in particularly implies that $w$ begins with an edge $T \leftarrow N$. By snipping cycles and appropriate shortcuts we obtain a definite-status, colliderless, non-causal path, where we use that $N$ will always be of definite status and therefore the directed edge into $T$ will never be removed. This path $p$ is blocked given $\mathbf{Z}$ and therefore contains a node in $\mathbf{Z}$. Therefore, so does $w$ which implies that it is blocked given $\mathbf{Z}$. Suppose now that $w$ contains colliders $C_1, \ldots, C_k$. By snipping cycles and taking shortcuts we can again obtain a definite-status path $p$ from $T$ to $Y$. Suppose $p$ is possibly directed, that is, consists of possibly causal nodes. Then at least on of the colliders must be a descendant of a possibly causal node and therefore $w$ is blocked. If $p$ is not possibly directed it must either contain a non-collider in $\mathbf{Z}$ that is also a non-collider on $w$ or a collider $C$, such that $\mathrm{De}(C, \mathcal{G}) \cap \mathbf{Z} \neq \emptyset$. In the former case, $w$ is obviously blocked. In the latter case, at least one of the $C_i$ must satisfy $C_i \notin \mathbf{Z}$ which again suffices to conclude that $w$ is blocked. $\qquad\square$

With this modified adjustment criterion we can algorithmically verify that a set $\mathbf{Z}$ is a valid adjustment set for $(\mathbf{T}, Y)$ in DAG or CPDAG $\mathcal{G}$, by verifying that

1. no proper possibly directed walk that does not begin with a directed edge out of $\mathbf{T}$ reaches $Y$, and
2. no proper possibly directed walk that contains a node in $\mathbf{Z}$ reaches $Y$, and
3. no proper definite-status non-causal walk that is not blocked by $\mathbf{Z}$ reaches $Y$.

Condition 1. is equivalent to Condition 1. in Lemma 22; since we need to verify that no such walk reaches $Y$, the reachability algorithm will need to continue walking possibly directed walks that do not start with an edge out of $\mathbf{T}$ even if they are blocked by $\mathbf{Z}$. Condition 2. is equivalent to Condition 2. in Lemma 22; since we need to verify that no such walk reaches $Y$, the reachability algorithm will need to continue walking possibly directed walks that start with an edge out of $\mathbf{T}$ even if they contain a node in $\mathbf{Z}$. Condition 3. is equivalent to the blocking Condition 3. in Lemma 22; since the (non-)existence of blocked non-causal paths does not appear in any of the conditions, the reachability algorithm can stop walking non-causal walks upon reaching a blocking node in $\mathbf{Z}$; however, the blocking condition poses a problem as verifying whether a path is blocked requires a non-local check to verify that it is of definite-status. In the following subsection, we show how to verify blocking in CPDAGs with a reachability algorithm while avoiding this non-local definite-status check.

## D.2   D-SEPARATION VIA A REACHABILITY ALGORITHM FOR CPDAGS

In this section, we show how to verify blocking in CPDAGs by a reachability algorithm without needing to discern the non-local property of a walk being definite-status or not. This is necessary to enable the use of reachability algorithms with local decision rules to verify the blocking condition on definite-status walks in the modified adjustment criterion. We show this in 5 steps:

- Lemma 23: Indefinite-status paths are irrelevant for d-separation in CPDAGs
- Lemma 24: Existence of open definite-status walks or paths coincides in CPDAGs
- Lemma 25: Reachability algorithm with non-local decision rules for d-connectedness in CPDAGs
- Lemma 26: We may treat some indefinite-status walks as definite-status
- Lemma 27: We may treat some more indefinite-status walks as definite-status

The following lemma by Henckel et al. [2022] characterizes d-separation in a CPDAG in terms of definite-status paths.

**Lemma 23** (Indefinite-status paths are irrelevant for d-separation in CPDAGs)**.** *Consider node sets* $\mathbf{T}, \mathbf{Y}$ *and* $\mathbf{Z}$ *in a CPDAG* $\mathcal{G}$*. Then* $\mathbf{T}$ *is d-separated from* $\mathbf{Y}$ *given* $\mathbf{Z}$ *in every DAG* $\mathcal{D} \in [\mathcal{G}]$ *if an only if every definite-status path from* $\mathbf{T}$ *to* $\mathbf{Y}$ *is blocked given* $\mathbf{Z}$ *in* $\mathcal{G}$*.*

Checking whether a collider is open on a definite-status path requires checking a non-local condition, as we need to consider all descendants of the collider. In DAGs we can circumvent that by considering walks instead, as it is possible to show that an open path exists if and only if an open walk exists. We now show that a similar result holds for CPDAGs, connecting definite-status paths and connecting definite-status walks.

**Lemma 24** (Existence of open definite-status walks or paths coincides in CPDAGs). *Consider node sets $\mathbf{T}, \mathbf{Y}$ and $\mathbf{Z}$ in a CPDAG $\mathcal{G}$. Then there exists a definite-status path from $\mathbf{T}$ to $\mathbf{Y}$ that is open given $\mathbf{Z}$ if and only if there exists a definite-status walk from $\mathbf{T}$ to $\mathbf{Y}$ that is open given $\mathbf{Z}$.*

*Proof.* Let $p$ be a definite-status path from some $T \in \mathbf{T}$ to some $Y \in \mathbf{Y}$ that is open given $\mathbf{Z}$ in $\mathcal{G}$. Let $C_1, \ldots, C_k$ be all colliders on $p$. By assumption there exist directed paths $q_1, \ldots, q_k$ from $C_i$ to some node $Z_i \in \mathbf{Z}$ that we choose to not contain any other node in $\mathbf{Z}$. Then $w = p(T, C_1) \oplus q_1(C_1, Z_1) \oplus q_1(Z_1, C_1) \oplus \cdots \oplus p(C_k, Y)$ is a definite-status walk from $T$ to $Y$ that is open given $\mathbf{Z}$.

For the converse direction consider a walk $w$ and let $I$ be the node closest to $T$ on $w$ that appears twice $w$, i.e., $w = w(T, I) \oplus w(I, I) \oplus w(I, Y)$. Consider the walk $w' = w(T, I) \oplus w(I, Y)$. We will now show that either $w'$ itself is a definite-status walk from $T$ to $Y$ such that no no-collider is in $\mathbf{Z}$ and every collider has a descendant in $\mathbf{Z}$ or that we can construct a shortcut walk that is. Since $w'$ contains at least one repeating node less than $w$ we can then iterate this contraction to obtain a definite-status path open given $\mathbf{Z}$. Every node on $w'$ inherits their definite status from the path $w$ except for $I$. Suppose $I \in \mathbf{Z}$, then $I$ must be a collider whenever it appears on $w$ and therefore it also a definite-status collider on $w'$ which is therefore of the claimed form. Suppose now that $I \notin \mathbf{Z}$. Then it must be a non-collider whenever it appears on $w$. There are three cases to consider: a) $I$ is a collider on, $w'$ b) $I$ is a definite-status non-collider on $w'$ and c) $I$ is not of definite status on $w'$. In case a) $w$ must have been of the form $T \cdots \to I \to \cdots \leftarrow I \leftarrow \cdots Y$. Therefore $w$ must contain a collider that is a descendant of $I$. Therefore $\mathrm{De}(I, \mathcal{G}) \cap \mathbf{Z} \neq \emptyset$. In case b) $w'$ trivially fulfills the required conditions.

In case c) we again consider three subcases: a) $A \to I - B$, b) $A - I \leftarrow B$ and c) $A - I - B$. In all three cases $A$ and $B$ are definite-status non-colliders on $w'$ as they inherited their status from $w$ and they cannot have been colliders. This also implies that $A, B \notin \mathbf{Z}$. In case a) there must also exist an edge $A \to B$ and we can replace $w'$ with $w'(T, A) \oplus w'(B, Y)$. The node $A$ is a definite non-collider on this new walk. If $B$ is also we are done. If it is not, i.e., we have the structure $A \to B - B'$ we can replace $w'$ again by repeating the argument we just made and taking the shortcut to $B'$. We can do so iteratively, until we encounter either a definite-status non-collider or $Y$ itself. Either way we obtain a definite-status walk such that every non-collider is not in $\mathbf{Z}$ and every collider has a descendant that is. Case b) follows by the exact same argument reversing the roles of $A$ and $B$. In case c) we must have an edge $A - B$. Again we replace $w'$ by taking the shortcut. If $A$ is not of definite status on the new walk, i.e, it contains the segment $A' - A - B$, there must exist an edge $A' - C$. We can again iteratively take shortcuts until we either obtain an $A'$ that is a definite-status non-collider or arrive at $T$. If $B$ is not of definite status we repeat the same procedure untile we arrive at definite-status $B'$ or $Y$. In all cases, we obtain a walk $w'$ of definite status that is open given $\mathbf{Z}$. $\qquad\square$

Based on Lemma 24 we can propose a reachability d-separation algorithm that additionally tracks and discerns whether a walk in a CPDAG is definite-status or not. The algorithm is a gensearch algorithm [Wienöbst et al., 2024, Algorithm 6] using the rule table given in Table 1 to traverse the graph. We now prove that this table is correct for d-separation in CPDAGs.

**Lemma 25** (Reachability algorithm with non-local decision rules for d-connectedness in CPDAGs.). *Consider a node set $\mathbf{T}$ in a CPDAG $\mathcal{G}$ and let $\mathbf{Z}$ be a node set in $\mathcal{G}$. The output of a reachability algorithm (gensearch by Wienöbst et al. [2024]) with the rule table given in Table 1 is the set of all nodes $Y \in \mathbf{V}$ that are d-connected with $\mathbf{T}$ given $\mathbf{Z}$ in $\mathcal{G}$.*

| case | continue to $W$ | yield $W$ |
|---|---|---|
| init $T - W$ | always | always |
| init $T \to W$ | always | always |
| init $T \leftarrow W$ | $W \notin \mathbf{Z}$ | always |
| $-V - W$ | $V \notin \mathbf{Z}$ and $V$ of definite status | $V \notin \mathbf{Z}$ and $V$ of definite status |
| $-V \to W$ | $V \notin \mathbf{Z}$ | $V \notin \mathbf{Z}$ |
| $-V \leftarrow W$ | never | never |
| $\to V - W$ | never | never |
| $\to V \to W$ | $V \notin \mathbf{Z}$ | $V \notin \mathbf{Z}$ |
| $\to V \leftarrow W$ | $V \in \mathbf{Z}$ | $V \in \mathbf{Z}$ |
| $\leftarrow V - W$ | $V \notin \mathbf{Z}$ | $V \notin \mathbf{Z}$ |
| $\leftarrow V \to W$ | $V \notin \mathbf{Z}$ | $V \notin \mathbf{Z}$ |
| $\leftarrow V \leftarrow W$ | $V \notin \mathbf{Z}$ | $V \notin \mathbf{Z}$ |

Table 1: Rule table for gensearch algorithm [Wienöbst et al., 2024] to compute all d-connected nodes in a CPDAG.

*Proof.* Every node adjacent to some $T \in \mathbf{T}$ is d-connected with $\mathbf{T}$ given $\mathbf{Z}$. Therefore, the initialization step of the reachability algorithm is correct. Now suppose that if we arrive at a node $V$ in the reachability algorithm that there exists some definite-status walk $w$ from some $T \in \mathbf{T}$ to $V$ that is open $\mathbf{Z}$ and consider a proper step of the reachability algorithm continuing from $V$. Based on the rule table we continue and yield $W$ precisely when appending the edge between $V$ and $W$ to $w$ results in a definite-status d-connecting walk $w'$ from $T$ to $W$, i.e., if $V$ is definite non-collider on $w'$ and $V \notin \mathbf{Z}$ or if $V$ is a collider on $w'$ and $V \in \mathbf{Z}$. By induction it follows that for every reachable node $Y$ there exists a definite-status d-connecting walk from $T$ which by Lemma 24 suffices to conclude that $\mathbf{T}$ and $Y$ are d-connected given $\mathbf{Z}$.

We now show that if a node $Y$ is d-connected with some $T \in \mathbf{T}$ given $\mathbf{Z}$ then it will be returned as reachable. By Lemma 24 there exist a definite-status d-connecting walk from $T$ to $Y$. We now make an induction argument on the length $l$ of the shortest such walk. If $l = 1$, the algorithm clearly returns $Y$, so suppose the algorithm returns all nodes with shortest paths of length $l = k - 1$ and suppose for $Y$ the shortest walk $w$ is of length $l = k$. Let $w'$ be the walk we obtain by removing the final edge from $w$. It's a walk of length $l$ that is d-connecting given $\mathbf{Z}$ and therefore this holds for it's end node $Y'$. By the induction hypothesis $Y'$ is reachable. Further, since $w$ is a definite-status d-connecting walk we can see that by applying the rule table to $Y'$ the algorithm will also return $Y$. $\qquad\square$

From an implementation perspective, it is problematic that in the fourth row of Table 1 we need to check whether $V$ is of definite status, since this requires storing adjacent nodes for previously visited nodes, that is, this rule is non-local. We now show that we can simply drop this check without modifying the output of the algorithm and in this way obtain a local algorithm.

**Lemma 26** (We may treat some indefinite-status walks as definite-status). *Consider a node set $\mathbf{T}$ in a CPDAG $\mathcal{G}$ and let $\mathbf{Z}$ be a node set in $\mathcal{G}$. Suppose we modify the rule table in Table 1 by treating the $-V-W$ case as if $V$ were of definite status (irrespective of whether it actually is). The resulting reachability algorithm has the same output as the original algorithm.*

*Proof.* The two algorithms agree locally in all cases except in the case $-V-W$, with $V \notin \mathbf{Z}$ not of definite status, where the original algorithm does not continue to $W$ and the modified algorithm does and also considers all vertices such that $W - W'$ or $W \to W'$ if $W \notin \mathbf{Z}$. Suppose that starting from some $T \in \mathbf{T}$, $V$ is the first node where the two algorithms disagree, i.e., there exists a $V'$ reachable by both algorithms such that $V' - V - W$, $V \notin \mathbf{Z}$ and $V' - W$. Since $V'$ is reachable with the original algorithm and we continue onto an undirected edge, $V' \notin \mathbf{Z}$ and we arrive at $V'$ via an edge of the form $\leftarrow V'$ or $-V$. The original algorithm will therefore reach $W$, either via $\leftarrow V' - W$ or $-V' - W$ unless in the latter case $V'$ is not of definite status. If the latter is the case we can repeat the argument to obtain a new $V'$ until we either arrive at a $V'$ that is of definite status or the walk $T - W$. In either case, $W$ is reachable and if $W \notin \mathbf{Z}$ we will consider all vertices such that $W - W'$ or $W \to W'$. $\qquad\square$

Finally, we will use a d-separation reachability algorithm within our new reachability algorithm to verify adjustment validity. Here, we also need to verify whether $\mathbf{Z}$ contains possibly causal nodes, that is, whether there exists a proper possibly directed path from $\mathbf{T}$ to $\mathbf{Y}$ that contains a node in $\mathbf{Z}$ (see Lemma 22). To do so, we may have to continue along segments of the form $\to N -$ if $N \notin \mathbf{Z}$ which are of indefinite status. We now show that we can further modify the d-separation rule tables to accommodate this without changing the output of the d-separation reachability algorithm. This will allow us to run all three checks required in the validity algorithm simultaneously. If we were only interested in d-separation, the rule table from Lemma 26 is more computationally efficient.

**Lemma 27** (We may treat some more indefinite-status walks as definite-status). *Consider a node set $\mathbf{T}$ in a CPDAG $\mathcal{G}$ and let $\mathbf{Z}$ be a node set in $\mathcal{G}$. Suppose we modify the rule table in Table 1 by treating the $-V-W$ case as if $V$ were of definite status and the $\to V -$ case by proceeding if $V \notin \mathbf{Z}$ (analogous to the rule for definite-status non-colliders). The resulting reachability algorithm has the same output as the original algorithm.*

*Proof.* We have already established in Lemma 26, that we can ignore the definite-status check without modifying the output so consider an algorithm based on this rule table and compare it to an algorithm with the additional rule modification stated in the lemma. The two algorithms agree locally in all cases except in the case $\to V - W$, with $V \notin \mathbf{Z}$, where the original algorithm does not continue to $W$ and the modified algorithm does and also considers all vertices such that $W - W'$ or $W \to W'$ if $W \notin \mathbf{Z}$. Suppose that starting from some $T \in \mathbf{T}$, $V$ is the first node where the two algorithms disagree, i.e., there exists a $V'$ reachable by both algorithms such that $V' \to V - W$, $V \notin \mathbf{Z}$ and $V' \to W$. Since $V'$ is reachable with the original algorithm and we continue onto an undirected edge, $V' \notin \mathbf{Z}$ and we arrive at $V'$ via an edge of the form $\leftarrow V'$ or $-V$. The original (and the modified) algorithm will therefore reach $W$, either via $\leftarrow V' \to W$ or $-V' \to W$. In either case,

$W$ is reachable and if $W \notin \mathbf{Z}$ we will consider all vertices such that $W - W'$ or $W \rightarrow W'$ and if $W \in \mathbf{Z}$ we will consider all vertices such that $W \leftarrow W'$. This means $W'$ is reachable either way and we will in fact move onto a larger number of the adjacent nodes of $W$, regardless. The extra check we make in the modified algorithm therefore does not modify the output. $\qquad\square$

## D.3 WALK-STATUS IN REACHABILITY ALGORITHMS

In addition to the result from Appendix D.1 and Appendix D.2, we require one more idea in order to be able to verify the adjustment criterion with a reachability algorithm: when verifying the adjustment criterion a walk may at first not violate the adjustment criterion but as we append edges to it it may become a walk whose existence violates the adjustment criterion. For example, a directed walk starting from $\mathbf{T}$ does not violate the adjustment criterion until it either encounters a node in $\mathbf{Z}$ or turns into an open definite-status non-causal walk. In order to track such walks, we need to carry forward information about the current walk's status when traversing the graph; specifically, we require that a quinary walk status be propagated forward. Knowing a walk's status allows us to use more complex local rules about when to continue a walk (for example, only stopping on a blocked walk when the walk is non-causal) and assigning different tags to a node depending on the status of the walk with which we reached it; two examples of this are a) adding a non-amenable tag to nodes we can reach with a possibly directed walk that begins with an undirected edge, and b) adding a not-validly-adjusted-for tag to nodes we can reach with a possibly directed walk that contains a node in $\mathbf{Z}$, since this is a walk that contains a possibly causal node.

Assume we start the algorithm in $\mathbf{T}$ given some $\mathbf{Z}$, by construction we never walk back into $\mathbf{T}$, that is, all walks are proper walks. Also, we never visit the same node via the same edge on a walk of the same type twice, such that our algorithm has runtime guarantee $O(p + m)$ analogous to the Bayes-Ball and gensearch algorithm. The walk status is quinary and one of the following:

$\mathbf{PD}^{T \rightarrow}\mathbf{OPEN}$ – These are possibly directed walks that started with an edge pointing out of $T$ are not blocked by $\mathbf{Z}$. Reaching a node $Y$ by a $\mathrm{PD}^{T \rightarrow}\mathrm{OPEN}$ walk does not tell us anything about whether $(\mathcal{G}, T, Y)$ is amenable or whether $\mathbf{Z}$ is a valid adjustment set for $(T, Y)$. Instead, we need to keep walking such a walk as it may turn into a blocked (possibly directed walk that started with an edge pointing out of $T$) walk upon passing through $\mathbf{Z}$ or a non-causal walk upon traversing along a backward-facing edge $\leftarrow$, which are walks that contain information about amenability or validity of adjustment for the nodes reached.

$\mathbf{PD}^{T -}\mathbf{OPEN}$ – These are possibly directed walks that started with an undirected edge out of $T$ and are not blocked by $\mathbf{Z}$. Reaching a node $Y$ by a $\mathrm{PD}^{T -}\mathrm{OPEN}$ walk tells us that $(\mathcal{G}, T, Y)$ is not amenable (which implies that $\mathbf{Z}$ cannot be a valid adjustment set for $(T, Y)$) as Condition 1 in the Modified Adjustment Criterion is violated. We need to keep walking such a walk as other nodes reached by it are also not amenable and as it may turn into a blocked (possibly directed walk that started with an undirected edge out of $T$) or a non-causal walk (which we need to check are blocked).

$\mathbf{PD}^{T \rightarrow}\mathbf{BLOCKED}$ – These are possibly directed walks that started with an edge pointing out of $T$ and contain a node in $\mathbf{Z}$. Reaching a node $Y$ by a $\mathrm{PD}^{T \rightarrow}\mathrm{BLOCKED}$ walk tells us that $\mathbf{Z}$ is not a valid adjustment set for $(T, Y)$ as Condition 2 in the Modified Adjustment Criterion is violated (the walk must have passed through a node in $\mathbf{Z}$). We need to keep walking such a walk as $\mathbf{Z}$ is also not a valid adjustment set for other nodes reached by this walk and as it may turn into a non-causal walk (which we need to check are blocked).

$\mathbf{PD}^{T -}\mathbf{BLOCKED}$ – These are possibly directed walks that started with an undirected edge out of $T$ and are blocked by $\mathbf{Z}$. Reaching a node $Y$ by a $\mathrm{PD}^{T -}\mathrm{BLOCKED}$ walk tells us that $(\mathcal{G}, T, Y)$ is not amenable (which implies that $\mathbf{Z}$ cannot be a valid adjustment set for $(T, Y)$) as Condition 1 in the Modified Adjustment Criterion is violated. We need to keep walking such a walk as other nodes reached by it are also not amenable and as it may turn into a non-causal walk (which we need to check are blocked).

$\mathbf{NONCAUSAL}$ — These are walks that have passed through at least one backward-facing edge and are thus non-causal and are not blocked by $\mathbf{Z}$; if they were blocked by $\mathbf{Z}$ we would just stop walking such a non-causal blocked walk. Reaching a node $Y$ by a $\mathrm{NONCAUSAL}$ walk tells us that $\mathbf{Z}$ is not a valid adjustment set for $(T, Y)$ as Condition 3 in the Modified Adjustment criterion is violated.

To summarise and help intuition, we provide the following illustration of the possible walk-status changes:

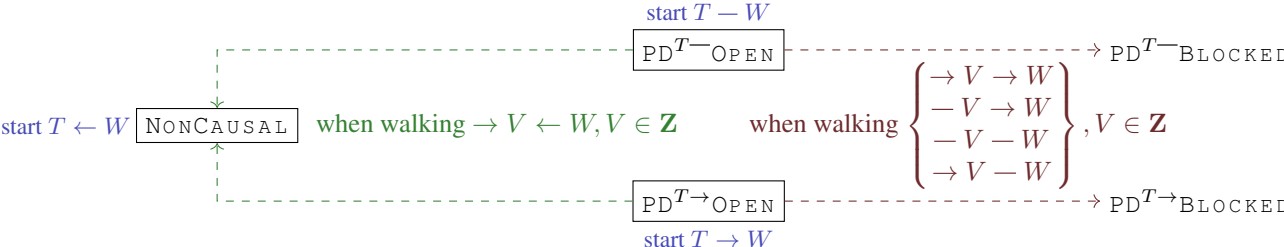

An instructive first example of a reachability algorithm is Algorithm 2 to check amenability, which we use also in our implementation of the identification strategies for CPDAGs. Here, the routine simplifies considerably, since we only start walking $\text{PD}^{T-}\text{OPEN}$ and $\text{PD}^{T-}\text{BLOCKED}$ walks from $T$ and all nodes $Y$ reached by a such a walk are nodes such that $(\mathcal{G}, T, Y)$ is not amenable.

Finally, in Algorithm 3 we present the key to efficiently implementing adjustment verification for our adjustment-based identification distances: Given a graph (DAG or CPDAG) $\mathcal{G}$, treatment $T$, and candidate adjustment set $\mathbf{Z}$ with $T \notin \mathbf{Z}$, Algorithm 3 returns in $O(p + m)$ time two lists a) NAM ("not amenable") containing all $Y \notin \mathbf{T}$ such that $(\mathcal{G}, T, Y)$ is not amenable, and b) NVA ("not validly adjusted for") containing all $Y \notin \mathbf{T} \cup \mathbf{Z}$ such that $\mathbf{Z}$ is not a valid adjustment set for $(T, Y)$ in $\mathcal{G}$ and all $Y \in \mathbf{Z}$.

---

**Algorithm 2** Check amenability of a CPDAG $\mathcal{G}$ relative to $(\mathbf{T}, Y)$ for a given set $\mathbf{T}$ of treatment nodes and all possible $Y$

1: **Input**: CPDAG $\mathcal{G}$ and a set of treatment nodes $\mathbf{T}$ in $\mathcal{G}$
2: **Output**: Set NAM of nodes $Y \notin \mathbf{T}$ in $\mathcal{G}$ such that $\mathcal{G}$ is **n**ot **am**enable relative to $(T, Y)$

3: **function** VISIT(arrivedby, $V$)
4:     visited.insert($V$)
5:     **if** arrivedby == init **then**     ▷ Start walking proper possibly directed walks that do not start out of $\mathbf{T}$
6:         **for** $W$ in AdjacentNodes($V$) $\setminus \mathbf{T}$ **do**
7:             **if** $W$ not in visited **then**
8:                 VISIT($-$, $W$)
9:     **else**                                ▷ Continue walking proper possibly directed walks
10:         NAM.push($V$)         ▷ Reached $V$ by a proper possibly directed walk that does not start out of $\mathbf{T}$
11:         **for** $W$ in AdjacentNodes($V$) $\setminus \mathbf{T}$ **do**
12:             **if** $W$ not in visited **then**
13:                 VISIT($-$, $W$)
14:         **for** $W$ in Ch($V$) $\setminus \mathbf{T}$ **do**
15:             **if** $W$ not in visited **then**
16:                 VISIT($\rightarrow$, $W$)

17: Initialise NAM as empty set
18: Initialise visited as empty HashSet

19: **for** $V$ in $\mathbf{T}$ **do** VISIT(init, $V$)

20: **return** NAM

---

**Algorithm 3** Validate $\mathbf{Z}$ as adjustment set relative to $(\mathbf{T}, Y)$ for a given set $\mathbf{T}$ of treatment nodes and all possible $Y$ in $\mathcal{G}$

1: **Input**: CPDAG (or DAG) $\mathcal{G}$, a set of treatment nodes $\mathbf{T}$ in $\mathcal{G}$, and a set of adjustment nodes $\mathbf{Z}$ in $\mathcal{G}$ with $\mathbf{T} \cap \mathbf{Z} = \emptyset$

2: **Output**: Set NAM of nodes $Y \notin \mathbf{T}$ in $\mathcal{G}$ such that $\mathcal{G}$ is **n**ot **am**enable relative to $(T, Y)$

3:          Set NVA of nodes $Y \notin \mathbf{T}$ in $\mathcal{G}$ such that $\mathbf{Z}$ is **n**ot a **v**alid **a**djustment set for $(T, Y)$ in $\mathcal{G}$

4: **function** NEXTSTEPS(`arrivedby`, $V$)                                        ▷ Return (`moveonby`, $W$, `blocked`) triplets

5:     Initialise `next` as empty set

6:     **if** `arrivedby` $==\; \rightarrow$ **then**

7:         **for** $W$ in $\mathrm{Pa}(V) \setminus \mathbf{T}$ **do**                            ▷ collider $\rightarrow V \leftarrow W$

8:             `next.push`$((\leftarrow,\; W,\; \mathbb{1}(V \notin \mathbf{Z})))$

9:     **else if** `arrivedby` in $\{\text{init}, \leftarrow\}$ **then**

10:        **for** $W$ in $\mathrm{Pa}(V) \setminus \mathbf{T}$ **do**                                   ▷ $\leftarrow V \leftarrow W$

11:            `next.push`$((\leftarrow,\; W,\; \mathbb{1}(V \in \mathbf{Z})))$

12:     **for** $W$ in $\mathrm{AdjacentNodes}(V) \setminus \mathbf{T}$ **do**         ▷ $\rightarrow V - W$ or $- V - W$ or $\leftarrow V - W$

13:        `next.push`$((-,\; W,\; \mathbb{1}(V \in \mathbf{Z})))$

14:     **for** $W$ in $\mathrm{Ch}(V) \setminus \mathbf{T}$ **do**                 ▷ $\rightarrow V \rightarrow W$ or $- V \rightarrow W$ or $\leftarrow V \rightarrow W$

15:        `next.push`$((\rightarrow,\; W,\; \mathbb{1}(V \in \mathbf{Z})))$

16:     **return** `next`                                        ▷ omits steps to $\mathbf{T}$ and $- V \leftarrow W$

17: **function** VISIT((`arrivedby`, $V$, `walkstatus`))

18:     `visited.insert`((`arrivedby`, $V$, `walkstatus`))

19:     **if** `walkstatus` in $\{\mathrm{PD}^{T-}\text{OPEN}, \mathrm{PD}^{T-}\text{BLOCKED}\}$ **then**

20:        `NAM.push`$(V)$ and `NVA.push`$(V)$   ▷ Reached $V$ by a possibly directed walk that does not start out of $\mathbf{T}$

21:     **else if** `walkstatus` $==$ NONCAUSAL **then**

22:        `NVA.push`$(V)$                            ▷ Reached $V$ by a non-causal walk that is not blocked by $\mathbf{Z}$

23:     **else if** `walkstatus` $==$ $\mathrm{PD}^{T\rightarrow}$BLOCKED **then**

24:        `NVA.push`$(V)$                          ▷ Reached $V$ by a possibly directed walk that is blocked by $\mathbf{Z}$

25:     **for** (`moveonby`, $W$, `blocked`) in NEXTSTEPS(`arrivedby`, $V$) **do**

26:        `next` = `none`

27:        **if** `walkstatus` $==$ init **then**

28:           **if** `moveonby` $==\; \rightarrow$ **then** `next` $= (\rightarrow,\; W,\; \mathrm{PD}^{T\rightarrow}\text{OPEN})$   ▷ Start possibly directed walk $\mathbf{T} \rightarrow$

29:           **else if** `moveonby` $==\; -$ **then** `next` $= (-,\; W,\; \mathrm{PD}^{T-}\text{OPEN})$ ▷ Start possibly directed walk $\mathbf{T} -$

30:           **else if** `moveonby` $==\; \leftarrow$ **then** `next` $= (\leftarrow,\; W,\; \text{NONCAUSAL})$           ▷ Start non-causal walk

31:        **else if** `walkstatus` in $\{\mathrm{PD}^{T\rightarrow}\text{OPEN}, \mathrm{PD}^{T\rightarrow}\text{BLOCKED}\}$ **then**

32:           **if** `moveonby` in $\{\rightarrow, -\}$ **then**

33:              **if** `blocked` $==$ false **then** `next` $= (\text{moveonby},\; W,\; \text{walkstatus})$

34:              **else if** `blocked` $==$ true **then** `next` $= (\text{moveonby},\; W,\; \mathrm{PD}^{T\rightarrow}\text{BLOCKED})$

35:           **else if** `moveonby` $==\; \leftarrow$ and `blocked` $==$ false and `walkstatus` $==$ $\mathrm{PD}^{T\rightarrow}$OPEN **then**

36:              `next` $= (\text{moveonby},\; W,\; \text{NONCAUSAL})$

37:        **else if** `walkstatus` in $\{\mathrm{PD}^{T-}\text{OPEN}, \mathrm{PD}^{T-}\text{BLOCKED}\}$ **then**

38:           **if** `moveonby` in $\{\rightarrow, -\}$ **then**

39:              **if** `blocked` $==$ false **then** `next` $= (\text{moveonby},\; W,\; \text{walkstatus})$

40:              **else if** `blocked` $==$ true **then** `next` $= (\text{moveonby},\; W,\; \mathrm{PD}^{T-}\text{BLOCKED})$

41:           **else if** `moveonby` $==\; \leftarrow$ and `blocked` $==$ false and `walkstatus` $==$ $\mathrm{PD}^{T-}$OPEN **then**

42:              `next` $= (\text{moveonby},\; W,\; \text{NONCAUSAL})$

43:        **else if** `walkstatus` $==$ NONCAUSAL and `blocked` $==$ false **then**

44:           `next` $= (\text{moveonby},\; W,\; \text{NONCAUSAL})$

45:        **if** `next` is not `none` and `next` not in `visited` **then** VISIT(`next`)

46: Initialise NAM as empty set

47: Initialise NVA=$\mathbf{Z}$

48: Initialise `visited` as empty HashSet

49: **for** $V$ in $\mathbf{T}$ **do** VISIT((init, $V$, init))

50: **return** NAM and NVA

## D.4 OUR ALGORITHM ENABLES EFFICIENT CALCULATION OF PARENT-, ANCESTOR-, AND OSET-AID

For the three distances, the Parent-AID, Ancestor-AID, and Oset-AID, we need to identify adjustment sets in $\mathcal{G}_{\text{guess}}$ with an additional amenability check in case $\mathcal{G}_{\text{guess}}$ is a CPDAG and then verify the proposed adjustment sets in $\mathcal{G}_{\text{true}}$. While algorithms with $O(p + m)$ runtime exist for each involved computation, the algorithmic development in the preceding subsections is crucial to enable efficient calculation of the distances: Algorithm 3 enables us to verify adjustment sets for all $\{(T, Y') \mid Y' \in \mathbf{V} \setminus \{T\}\}$ with one $O(p + m)$ run, instead of performing $(p - 1)$ separate runs of a valid adjustment verifier algorithm. For simplicity and as an instructive example, we first discuss our implementation of the Parent-AID for DAGs and then present the general routine for implementing our distances.

### D.4.1 Calculating the Parent Adjustment Identification Distance Efficiently

To calculate the Parent-AID between two DAGs $\mathcal{G}_{\text{true}}$ and $\mathcal{G}_{\text{guess}}$ over $p$ nodes and $m$ edges, we need to iterate over all tuples $(T, Y)$ of nodes, obtain the parent set of the treatment in $\mathcal{G}_{\text{guess}}$, and check whether this set is a valid adjustment set in $\mathcal{G}_{\text{true}}$ with respect to $(T, Y)$. For their SID implementation, Peters and Bühlmann [2015] report a worst-case runtime of $O(p \cdot \log_2(p) \cdot p^3)$ where the factor $p^3$ corresponds to squaring of the adjacency matrix of $\mathcal{G}_{\text{true}}$ which is done $\lceil \log_2(p) \rceil$ times to assemble a path matrix that codes which nodes are reachable from each of the $n$ treatment nodes.[1]

Combining the above algorithms, we can calculate the Parent-AID with an algorithm with runtime $O(p(p + m))$ as follows:

- Initialise the mistake count $c = 0$
- For each node $T$ (each of the following steps can be completed in $O(p + m)$ time)
    - Obtain $\mathbf{Z}$ as the set of parents of $T$ in $\mathcal{G}_{\text{guess}}$
    - Obtain $\mathbf{ND}$ as the set of non-descendants of $T$ in $\mathcal{G}_{\text{true}}$
    - Obtain $\mathbf{NVA}$ as the the set of nodes $Y$ such that $\mathbf{Z}$ is not a valid adjustment set for $(T, Y)$ in $\mathcal{G}_{\text{true}}$
    - Add

$$\underbrace{|\mathbf{Z} \setminus \mathbf{ND}|}_{\text{guessed no effect, but descendant in } \mathcal{G}_{\text{true}}} + \underbrace{|\mathbf{Z}^{\complement} \cap \mathbf{NVA}|}_{\mathbf{Z} \text{ valid adjustment set in } \mathcal{G}_{\text{guess}}, \text{ but not in } \mathcal{G}_{\text{true}}}$$

    to the mistake count $c$
- Return $d^{\mathcal{I}_P}(\mathcal{G}_{\text{true}}, \mathcal{G}_{\text{guess}}) = c$

Our Parent-AID coincides with the SID only as distance between DAGs, but, in contrast to the SID, generalizes to CPDAGs. The multi-set SID between CPDAGs requires exponential runtime, while the Parent-AID between CPDAGs is still $O(p(p + m))$ as shown in the next subsection.

### D.4.2 Calculating Adjustment Identification Distances Efficiently

We fix the treatment $T$ and apply our algorithm to all tuples $(T, Y), T \neq Y$ simultaneously. We also group our identifying formulas as follow: For each $T$, the identification strategy algorithm returns a vector of (node, identifying formula) tuples which we code as a triple $(\mathbf{A}, \mathbf{B}, (Y, \mathbf{Z}(Y))_{Y \in \mathbf{C}})$ consisting of a) the set nodes $\mathbf{A}$ for which the causal effect from $T$ is not identifiable, b) a set of nodes $\mathbf{B}$ for which the causal effect from $T$ is zero and c) a set of two-tuples consisting of the remaining nodes $Y \in \mathbf{C} = (\mathbf{A} \cup \mathbf{B})^c$ and for each $Y$ a corresponding valid adjustment set $\mathbf{Z}(Y)$. To compute this vector, we first apply a reachability algorithm to compute $\mathbf{I} = \text{amen}(T, \mathcal{G}_{\text{guess}})$ (Algorithm 2), where $\text{amen}(T, \mathcal{G})$ denotes the set of nodes $Y$ such that $(\mathcal{G}, T, Y)$ is amenable. For the parent strategy we then compute $\mathbf{P} = \text{Pa}(T, \mathcal{G}_{\text{guess}})$ and return $(\mathbf{I}^c, \mathbf{P}, (Y, \mathbf{P})_{Y \in \mathbf{I} \setminus \mathbf{P}})$. For the ancestor strategy we compute $\mathbf{A} = \text{An}(T, \mathcal{G}), \mathbf{D} = \text{De}(T, \mathcal{G})$ and return $(\mathbf{I}^c, \mathbf{D}^c \cap \mathbf{I}, (N, \mathbf{A})_{N \in \mathbf{D}})$. For the Oset strategy, we compute $\mathbf{D}, \mathbf{O}(T, Y, \mathcal{G}) = \text{Pa}(\mathbf{D} \cap \text{An}(Y, \mathcal{G})) \setminus \mathbf{D}$ for each $Y \in \mathbf{D}$ and return $(\mathbf{I}^c, \mathbf{D}^c \cap \mathbf{I}, (Y, \mathbf{O}(T, Y, \mathcal{G}_{\text{guess}}))_{Y \in \mathbf{D}})$. We repeat these steps for each $T$ and return a vector of three-tuples. The overall complexity is therefore $O(p(p + m))$ for the parent and ancestor strategies, and $O(p^2(p + m))$ for the Oset strategy. The additional $p$ is due to $\mathbf{O}(T, Y, \mathcal{G})$ depending on $Y$, whereas $\text{Pa}(T, \mathcal{G})$ and $\text{An}(T, \mathcal{G})$ do not depend on $Y$.

---

[1]One may be able to reduce the cubic runtime for the matrix multiplication if the adjacency matrices exhibit extra known structure, though, the algorithm with the best known asymptotic runtime to date of $O(p^{2.37})$ is a galactic algorithm and not usable in practice [Alman and Williams, 2020]. For certain adjacency matrices, the Strassen algorithm for matrix multiplication may enable a reduction to $O(p^{\log_2(7)}) \approx O(p^{2.8})$.

Consider now the verification step for a fixed treatment $T$ and the corresponding triple $(\mathbf{A}, \mathbf{B}, (Y, \mathbf{Z}(Y))_{Y \in \mathbf{C}})$. We compute $\mathbf{I}' = \text{amen}(T, \mathcal{G}_{\text{true}})$ and $\mathbf{D}' = \text{De}(Y, \mathcal{G}_{\text{true}})$ in $O(p + m)$ using reachability algorithms. For each unique $\mathbf{Z}(Y)$ we then apply a reachability algorithm to compute set of nodes $N \in \mathbf{V}(T, \mathbf{Z}(Y), \mathcal{G}_{\text{true}})$ such that $\mathbf{Z}(Y)$ is a valid adjustment set relative to $(T, N)$ in $\mathcal{G}_{\text{true}}$ (Algorithm 3). For the parent and ancestor strategy there is only one $\mathbf{Z}(Y)$, so we only need to do this step once. We then add

$$|\mathbf{A} \cap \mathbf{I}'| + |\mathbf{B} \cap \mathbf{D}'| + \#\{Y \in \mathbf{C} \setminus \mathbf{V}(T, \mathbf{Z}(Y), \mathcal{G}_{\text{true}})\}$$

to the distance between $\mathcal{G}_{\text{true}}$ and $\mathcal{G}_{\text{guess}}$. As we have to repeat this for each $T$, the overall runtime for the verifier is $O(p(p + m))$ for the parent and ancestor strategies, and $O(p^2(p + m))$ for the Oset strategy.

# E   EMPIRICAL ANALYSIS OF ALGORITHMIC TIME COMPLEXITY

To empirically analyze the runtime complexity of our distance implementations, we evaluate the algorithms on inputs of varying size $p$ and measure the runtime $r_{\text{emp}}(p)$ (here, we use the wall time). For a given complexity, such as $O(p^2)$, we then project the runtime we would expect for any $p$ based on the runtime observed for the smallest size $p$. We denote the projected runtime for $p$ as $r_{\text{proj}}(p)$. The idea is that the ratio of the projected runtime $r_{\text{proj}}(p)$ and the observed empirical runtime $r_{\text{emp}}(p)$ approaches 1 in the limit of $p \to \infty$ if the implementation has the complexity used to compute $r_{\text{proj}}(p)$. For a given algorithm $\texttt{distance}(\mathcal{G}_{\text{true}}, \mathcal{G}_{\text{guess}})$ we proceed as follows.

- *(Grid of graph sizes)*
  Specify a grid of graph sizes $P$, for example, $P = (8, 16, 32, 64, 128, 256, 512, 1024)$.

- *(Observed empirical runtimes)*
  Sample, for each $p \in P$, 11 pairs of DAGs $\mathcal{G}_{\text{true}} = (\mathbf{V}, \mathbf{E}_{\text{true}})$ and $\mathcal{G}_{\text{guess}} = (\mathbf{V}, \mathbf{E}_{\text{guess}})$ with $|\mathbf{V}| = p$ and edges drawn independently with probability $20/(p - 1)$ for sparse and $0.3$ for dense graphs (the expected number of edges is thus linear in the number of nodes for sparse, and quadratic for dense graphs). We run $\texttt{distance}$ on those 11 pairs and compute the average runtime over these 11 runs, denoted $r_{\text{emp}}(p)$.

- *(Project the runtime under the given time complexity $O(c(p))$ where, for example $c(p) = p^2$)*
  We obtain the projected runtime for inputs of size $p$ under the given time complexity based on the smallest input size $p^* = \min(P)$ as

$$\text{projected runtime:} \quad r_{\text{proj}}(p) = c(p) \frac{r_{\text{emp}}(p^*)}{c(p^*)}.$$

- *(Relative projected runtime)*
  To compare the projected runtime to the observed empirical runtime, we then visualize the projected runtime as a fraction of the observed empirical runtime

$$\text{relative projected runtime:} \quad \frac{r_{\text{proj}}(p)}{r_{\text{emp}}(p)}.$$

As we consider asymptotic complexity, we need to evaluate for large enough $p$ to assess the following trends of relative projected runtimes for increasing $p$. If, when comparing to $O(c(p))$, the relative projected runtime increases with the number of nodes $p$, this indicates that the empirical time complexity is lower than $O(c(p))$. If, when comparing to $O(c(p))$, the relative projected runtime decreases with the number of nodes $p$, this indicates that the empirical time complexity is larger than $O(c(p))$. If the algorithm has time complexity $O(c(p))$ then we expect the relative projected runtime for this complexity to be close to constant.

# F   DISTANCE COMPARISON

The simulation study described in Section 7.2 is part of a larger study in which we consider 8 parameter settings. Specifically, we consider all possible combinations of the following three choices: i) $\mathcal{G}_{\text{true}}$ is a dense, respectively sparse graph randomly drawn as described in Section 7.1, ii) $\mathcal{G}_{\text{guess}}$ is $\mathcal{G}_{\text{true}}$ with one edge randomly removed, respectively $\mathcal{G}_{\text{guess}}$ is randomly drawn in the same way as $\mathcal{G}_{\text{true}}$ and iii) $\mathcal{G}_{\text{true}}$ and $\mathcal{G}_{\text{guess}}$ are 30-node, respectively 100-node graphs. For each of these 8 settings we randomly draw 300 pairs of $\mathcal{G}_{\text{true}}$ and $\mathcal{G}_{\text{guess}}$ graphs. For each pair we compute the three proposed adjustment identification distances and the SHD. To summarise the results we obtain correlation tables across the distances for each of the 8 experiments. We also provide corresponding scatter plots; to reduce clutter we do so only for the 30-node graphs.

## F.1  DISTANCES BETWEEN A RANDOM GRAPH AND A GRAPH WITH ONE EDGE REMOVED

Table 2 contains correlation tables for the setting that $\mathcal{G}_{\text{true}}$ is a random dense graph and $\mathcal{G}_{\text{guess}}$ is $\mathcal{G}_{\text{true}}$ with one edge removed as well as the average value for each distance over the 300 pairs. The left table contains the correlations for the $p = 30$ case and the right those for the $p = 100$ case. We also provide a corresponding scatter plot for the $p = 30$ case in Figure 5. We do not include the SHD, as the SHD between $\mathcal{G}_{\text{true}}$ and $\mathcal{G}_{\text{guess}}$ is by construction 1.

When $\mathcal{G}_{\text{guess}}$ graphs are close to the true graphs $\mathcal{G}_{\text{true}}$ the correlation between the three distances is surprisingly small; particularly the correlations between the Oset-AID, respectively the Ancestor-AID and the Parent-AID are small. This may be explained by how the Parent-AID treats node tuples $(T, Y)$ for which $Y \notin \text{De}(T, \mathcal{G}_{\text{guess}})$ differently from the Ancestor- and Oset-AID: For such tuples, both the Oset and ancestor adjustment strategy return $f(y)$, whereas parent adjustment only does so if $Y \in \text{Pa}(T, \mathcal{G}_{\text{guess}})$. As a result, small differences between $\mathcal{G}_{\text{true}}$ and $\mathcal{G}_{\text{guess}}$ tend to lead to a larger number of mistakes if we apply parent adjustment as opposed to Oset or ancestor adjustment. This is also reflected in the larger average of the Parent-AID. Another interesting pattern visible in the scatter plot, is the number of cases where a large Parent-AID coincides with a small or even zero Ancestor-AID. This illustrates how $\mathcal{G}_{\text{guess}}$ graphs that respect the causal orders of $\mathcal{G}_{\text{true}}$ may nonetheless be deemed very distant by the Parent-AID. Overall, the results indicate that the three distances are meaningfully different, that is, they capture distinct information.

| $p = 30$ | Ancestor-AID | Oset-AID | Parent-AID |
|---|---|---|---|
| Ancestor-AID | 1 | 0.7281 | 0.0886 |
| Oset-AID | 0.7281 | 1 | 0.2080 |
| Parent-AID | 0.0886 | 0.2080 | 1 |
| Average dist. | 2.0 | 5.9 | 11.2 |

| $p = 100$ | Ancestor-AID | Oset-AID | Parent-AID |
|---|---|---|---|
| Ancestor-AID | 1 | 0.7441 | 0.3717 |
| Oset-AID | 0.7441 | 1 | 0.3114 |
| Parent-AID | 0.3717 | 0.3114 | 1 |
| Average dist. | 3.4 | 13.6 | 39.8 |

Table 2: Correlation tables for the case that $\mathcal{G}_{\text{true}}$ is a random dense graph and $\mathcal{G}_{\text{guess}}$ is $\mathcal{G}_{\text{true}}$ with one edge removed.

Table 3 contains correlation tables for the setting that $\mathcal{G}_{\text{true}}$ is a random sparse graph and $\mathcal{G}_{\text{guess}}$ is $\mathcal{G}_{\text{true}}$ with one edge removed as well as the average value for each distance over the 300 pairs. The left table contains the correlations for the $p = 30$ case and the right those for the $p = 100$ case. We also provide a corresponding scatter plot for the $p = 30$ case in Figure 6. We do not include the SHD, as the SHD between $\mathcal{G}_{\text{true}}$ and $\mathcal{G}_{\text{guess}}$ is by construction 1.

The results are overall qualitatively similar to the results for dense graphs, which indicates that the distinct characteristics of the three distances are not specific to sparse or small graphs but are in fact a characteristic of the three distances.

| $p = 30$ | Ancestor-AID | Oset-AID | Parent-AID |
|---|---|---|---|
| Ancestor-AID | 1 | 0.8564 | 0.4377 |
| Oset-AID | 0.8564 | 1 | 0.3749 |
| Parent-AID | 0.4377 | 0.3749 | 1 |
| Average dist. | 1.0 | 2.4 | 8.3 |

| $p = 100$ | Ancestor-AID | Oset-AID | Parent-AID |
|---|---|---|---|
| Ancestor-AID | 1 | 0.7280 | 0.3019 |
| Oset-AID | 0.7280 | 1 | 0.2685 |
| Parent-AID | 0.3019 | 0.2685 | 1 |
| Average dist. | 4.1 | 24.8 | 42.7 |

Table 3: Correlation tables for the case that $\mathcal{G}_{\text{true}}$ is a random sparse graph and $\mathcal{G}_{\text{guess}}$ is $\mathcal{G}_{\text{true}}$ with one edge removed.

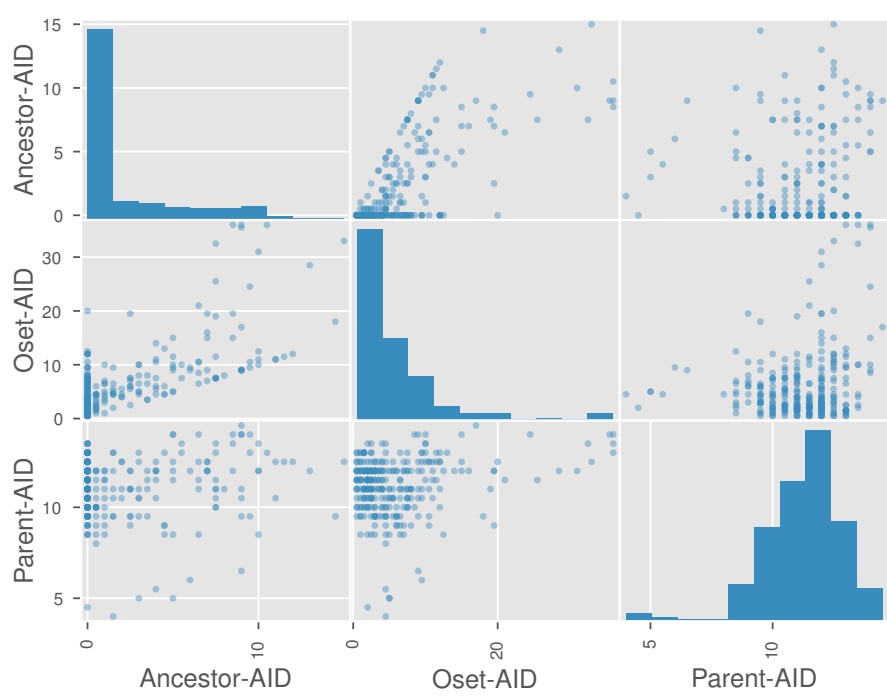

Figure 5: Scatter plot for the case that $\mathcal{G}_{\text{true}}$ is a random 30-node dense graph and $\mathcal{G}_{\text{guess}}$ is $\mathcal{G}_{\text{true}}$ with one edge removed.

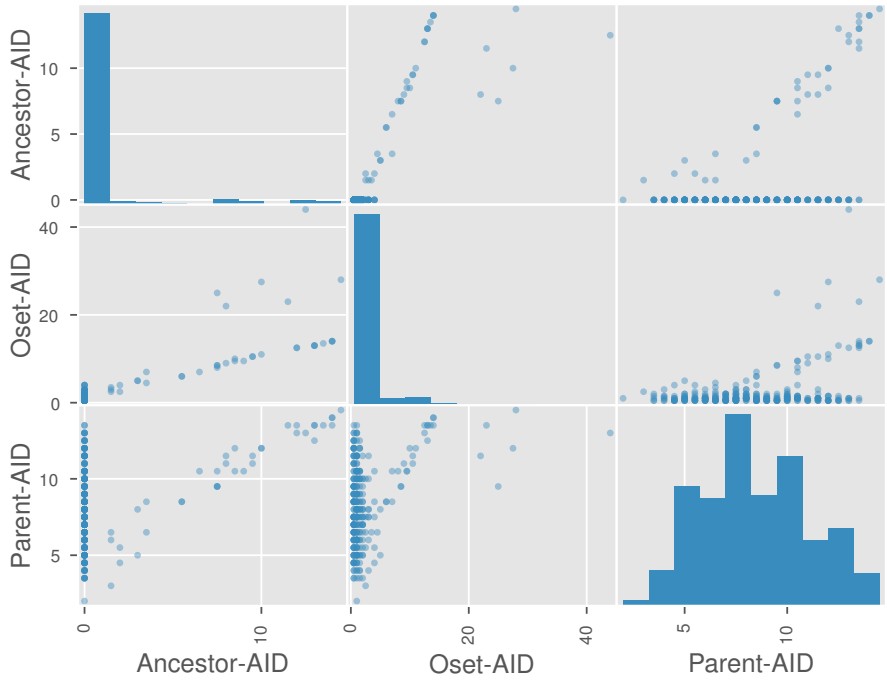

Figure 6: Scatter plot for the case that $\mathcal{G}_{\text{true}}$ is a random 30-node sparse graph and $\mathcal{G}_{\text{guess}}$ is $\mathcal{G}_{\text{true}}$ with one edge removed.

## F.2    DISTANCES BETWEEN TWO RANDOM GRAPHS

Table 4 contains correlation tables for the setting that $\mathcal{G}_{\text{true}}$ and $\mathcal{G}_{\text{guess}}$ are random dense graphs. The left table contains the correlations for the $p = 30$ case and the right those for the $p = 100$ case. We also provide a corresponding scatter plot for the $p = 30$ case in Figure 7.

When comparing two independently drawn random dense graphs, the distances are more strongly correlated than what was observed in Appendix F.1. Especially, the Ancestor-AID and the Oset-AID are very strongly correlated. Overall, the results indicate that, while the distances have distinct characteristics, the practical difference may be less relevant when comparing vastly different graphs as opposed to close-by graphs as in Appendix F.1.

| $p = 30$ | Ancestor-AID | Oset-AID | Parent-AID | SHD |
|---|---|---|---|---|
| Ancestor-AID | 1 | 0.9474 | 0.7429 | 0.4673 |
| Oset-AID | 0.9474 | 1 | 0.5715 | 0.5233 |
| Parent-AID | 0.7429 | 0.5715 | 1 | 0.1769 |
| SHD | 0.4673 | 0.5233 | 0.1769 | 1 |
| Average dist. | 253.6 | 258.7 | 383.0 | 202.4 |

| $p = 100$ | Ancestor-AID | Oset-AID | Parent-AID | SHD |
|---|---|---|---|---|
| Ancestor-AID | 1 | 0.9819 | 0.8512 | 0.4038 |
| Oset-AID | 0.9819 | 1 | 0.7704 | 0.3838 |
| Parent-AID | 0.8512 | 0.7704 | 1 | 0.2959 |
| SHD | 0.4038 | 0.3838 | 0.2959 | 1 |
| Average dist. | 3469.8 | 3475.6 | 4539.0 | 2299.7 |

Table 4: Correlation tables for the case that $\mathcal{G}_{\text{true}}$ and $\mathcal{G}_{\text{guess}}$ are random dense graphs.

Table 5 contains correlation tables for the setting that $\mathcal{G}_{\text{true}}$ and $\mathcal{G}_{\text{guess}}$ are random sparse graphs. The left table contains the correlations for the $p = 30$ case and the right those for the $p = 100$ case. We also provide a corresponding scatter plot for the $p = 30$ case in Figure 8.

The results for sparse graphs are overall qualitatively similar to the results for dense graphs. One notable difference is the scatter plot between the Ancestor-AID and the Oset-AID which shows that in many cases the Ancestor-AID and the Oset-AID are the same but that in the cases where they differ they are rarely just slightly different. We are uncertain what drives this behavior and why it is less pronounced in dense graphs. One potential explanation is that the Ancestor-AID and the Oset-AID count a mistake whenever two variables are in ancestral relationship in $\mathcal{G}_{\text{guess}}$ but not in $\mathcal{G}_{\text{true}}$ or vice versa. Similarly, they do not count a mistake when two variables that are not in ancestral relationship in $\mathcal{G}_{\text{guess}}$ also are not in ancestral relationship in $\mathcal{G}_{\text{true}}$. When comparing two random sparse graphs, this behavior may cover the large majority of node pairs and therefore the Ancestor-AID and the Oset-AID between random sparse graphs are often very similar. The distances only possibly differ for node pairs $(T, Y)$ where $T$ is ancestor of $Y$ in both graphs and only one of the two adjustment strategies applied to $\mathcal{G}_{\text{guess}}$ yields an adjustment set that is also a valid adjustment set in $\mathcal{G}_{\text{true}}$.

| $p = 30$ | Ancestor-AID | Oset-AID | Parent-AID | SHD |
|---|---|---|---|---|
| Ancestor-AID | 1 | 0.9782 | 0.9390 | 0.8170 |
| Oset-AID | 0.9782 | 1 | 0.9000 | 0.8264 |
| Parent-AID | 0.9390 | 0.9000 | 1 | 0.8026 |
| SHD | 0.8170 | 0.8264 | 0.8026 | 1 |
| Average dist. | 316.0 | 316.8 | 356.0 | 289.2 |

| $p = 100$ | Ancestor-AID | Oset-AID | Parent-AID | SHD |
|---|---|---|---|---|
| Ancestor-AID | 1 | 0.9668 | 0.6965 | 0.3124 |
| Oset-AID | 0.9668 | 1 | 0.5212 | 0.2914 |
| Parent-AID | 0.6965 | 0.5212 | 1 | 0.1324 |
| SHD | 0.3124 | 0.2914 | 0.1324 | 1 |
| Average dist. | 3229.3 | 3242.9 | 4639.4 | 1694.5 |

Table 5: Correlation tables for the case that $\mathcal{G}_{\text{true}}$ and $\mathcal{G}_{\text{guess}}$ are random sparse graphs.

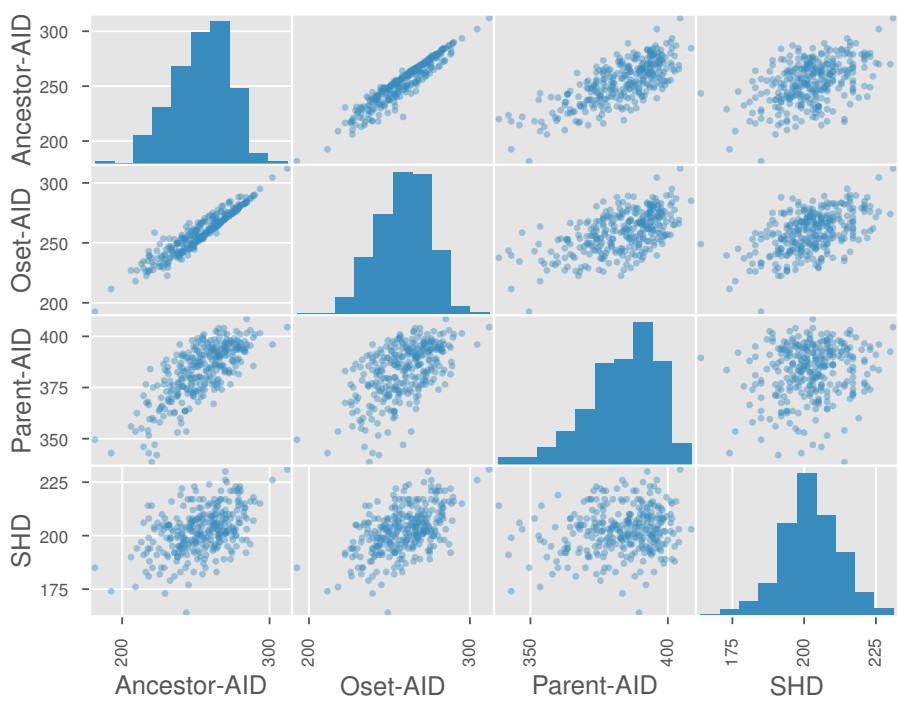

Figure 7: Scatter plot for the case that $\mathcal{G}_{\text{true}}$ and $\mathcal{G}_{\text{guess}}$ are random dense graphs with 30 nodes.

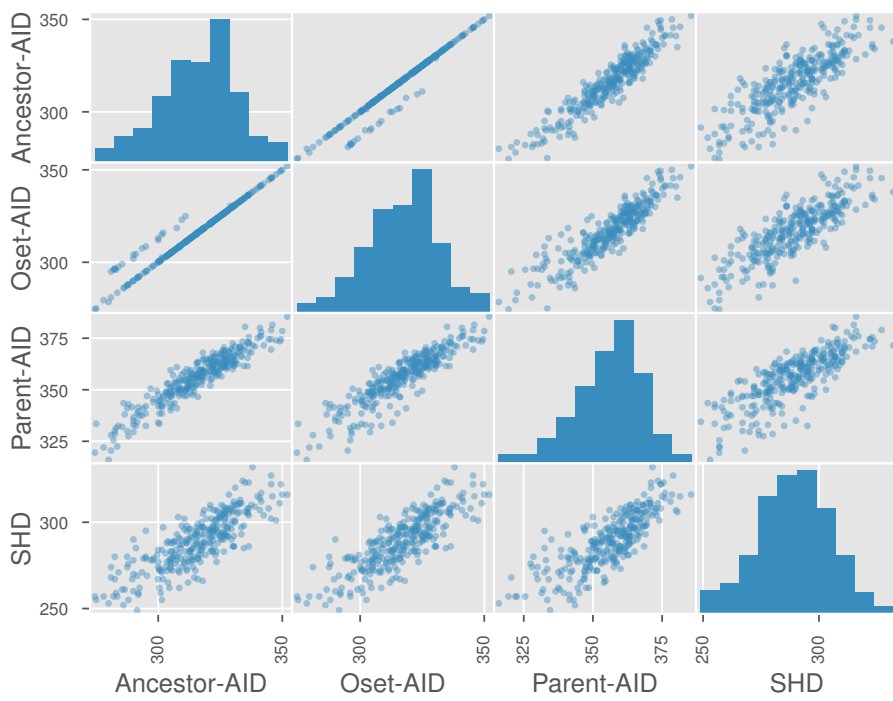

Figure 8: Scatter plot for the case that $\mathcal{G}_{\text{true}}$ and $\mathcal{G}_{\text{guess}}$ are random sparse graphs with 30 nodes.