# OpenReview forum: "Adjustment Identification Distance: A gadjid for Causal Structure Learning"
_auai.org/UAI/2024/Conference — UAI 2024 poster_

### Official Review · Reviewer_WDco · 2024-03-13

**Q2-1 Originality-Novelty:** 3
**Q2-2 Correctness-Technical Quality:** 3
**Q2-5 Clarity Of Writing:** 3

**Q10 Ethical Concerns:**

No.

**Q1 Summary And Contributions:**

In this work, the authors present a new family of causal distance measures for DAGs and CPDAGs that includes the Structural Intervention Distance as a special case.  They also provide an implementation of their Identification Distances with low time complexity. In particular, this significantly improves upon existing implementations of the SID.

**Q2-3 Extent To Which Claims Are Supported By Evidence:**

3: Good: the main claims are supported by convincing evidence (in the form of adequate experimental evaluation, proofs, (pseudo-)code, references, assumptions).

**Q2-4 Reproducibility:**

4: Excellent: key resources (e.g. proofs, code, data) are available and key details (e.g. proof sketches, experimental setup) are comprehensively described for competent researchers to confidently and easily reproduce the main results.

**Q3 Main Strengths:**

The paper is well-written and the introduction of new comparison measures for causal graphs that are of a causal nature is a welcome and needed contribution to the field. The measures presented here (in particular, the Ancestor AID) have some desirable properties including low-time complexity. In my opinion, they are particularly well-suited for evaluating causal discovery methods when their output is to be used for effect estimation queries later on.

**Q4 Main Weakness:**

What I miss is an empirical analysis of the effective difference of the AIDs and other metrics. That is, it would be interesting to see, how differently an example method performs with respect to your AIDs and other metrics such as SHD on some example data. Such examples could also illustrate how much a performance evaluation is influenced by the choice of whether to compare DAGs or CPDAGs. I understand that a full-on performance analysis of a method is beyond the scope of this work, but right now, one can’t help but wonder how much your measures differ in practice. I am also less convinced by the adaptation to CPDAGs which warrants more discussion than the paper is currently providing, see detailed comments below.

**Q5 Detailed Comments To The Authors:**

•	P.1 before last paragraph on the right. Typo: is not a performance metric […]

•	P.3, first paragraph. The terminology here is a bit awkward in my opinion. You speak of compatibility of a probability density $f$ and a DAG, but then this property actually refers to multiple densities $(f(.| do(t)))_{t \in range(T)}$.

•	P4., first paragraph: This might be a good place to point out that your distances, like the SID, are not symmetric in their arguments.

•	P.4: second to last line: typo: lear

•	P.5.: I like your idea of having a measure of distance that depends on the causal order alone. However, aren’t there simpler ways of achieving this that would even give you proper metrics on the set of causal orders? For instance, one could define the distance between two causal orders as the number of transpositions that are needed to permute one order into the other, essentially employing the standard metric on the symmetric group. How would that compare to your approach?

•	P.6.: Prop. 16: Amenability already has a specific meaning in graph theory, so I would suggest to use a different term here.

•	P.6.: left side, last paragraph. Typo: CPADG

•	P.6: CPDAG Distances: do your distances still have similar properties like the one of Proposition 7 on CPDAGs? Is there any relationship between an identification distance between two DAGs and the analogous distance between their CPDAGs?

•	How ‘well-behaved’ are AIDs for CPDAGs if no (or only very few) effects are identifiable in the true CPDAG? In this case, it seems like an inferred CPDAG can have distance zero to the true one, even if many edges are mistakenly deleted, as long as nothing becomes identifiable. This does not seem particularly desirable.

•	P.7, first paragraph: given the shortcomings that you are describing here, I am not so convinced that cross-type comparisons are a good idea to compare different algorithms. If the distance between a DAG and its own CPDAG is not zero or at least minimal among all CPDAGs, it is not so clear what this comparison is even supposed to express. The alternative that you suggest in the next paragraph seems more meaningful to me.

•	P.7, second paragraph: the comparison between GES and PCALG mentioned here, brings the additional problem that the outputs of PCALG may not be CPDAGs at all due to finite sample conflicts, see e.g. the related work [Wahl and Runge 24, Metrics on Markov Equivalence Classes for Evaluating Causal Discovery Algorithms] and in this case computing your distance may be difficult. One can hardly blame your metric for this failure mode of constraint-based methods, but it might be worth mentioning here.

**Q9 Complying With Reviewing Instructions:**

Yes

---

> ### Author Rebuttal · Authors · 2024-04-04
>
> Dear Reviewer WDco,
>
> Thank you for the constructive feedback.
>
> We first respond to your 2 main questions:
>
>
> #### Differences between distances
>
> We agree that a simulation study that investigates how different causal discovery algorithms perform in terms of those metrics is an interesting avenue for future research (and likewise hope that future benchmark studies include additional distances for a more complete picture) but consider it beyond the scope of this paper, in which we develop and contribute a general framework and new algorithms for causal graph distances.
>
> We do agree, that illustrating empirically that the distances differ, is a valuable addition that will improve the paper (we will use the additional space in the camera-ready version for including the below). We ran two experiments to compare the distances:
>
> __Experiment A__: We draw 300 pairs (Gtrue, Gguess) of random dense graphs (30 nodes, edge probability 0.3, on average 130.5 edges). The distances calculated on those 300 graph-pairs differ and we obtain the following correlation matrix:
>
> |              |   ancestor_aid |   oset_aid |   parent_aid |      shd |
> |:-------------|---------------:|-----------:|-------------:|---------:|
> | ancestor_aid |       1        |   0.9474   |     0.7429   | 0.4673   |
> | oset_aid     |       0.9474   |   1        |     0.5715   | 0.5232   |
> | parent_aid   |       0.7429   |   0.5715   |     1        | 0.1769   |
> | shd          |       0.4673   |   0.5232   |     0.1769   | 1        |
>
>
> __Experiment B__: See the general response (which also reports mean distances).
>
>
> __The above results__, and additional settings plus scatter plots which we will include in the appendix of the camera-ready version, illustrate that the distances measure distinct information. Average distances can be efficiently calculated over several hundreds of graph-pairs to establish a reference value for any given graph distribution used in a causal discovery benchmarking study.
>
>
>
> #### Distance between CPDAGs
>
> Oftentimes all one can hope for when using a causal discovery algorithm is the identification of an equivalence class of the causal graph, i.e., its CPDAG. The information learnt and encoded in the CPDAG may be sufficient to identify some causal effects but not others, that is, even though a CPDAG has some undirected edges certain causal effects are identifiable knowing the equivalence class alone. In such case, we take a practitioner-perspective: a practitioner would use the CPDAG only to identify those causal effects that are identifiable (cf. amenability) but would refrain from doing so for pairs (T,Y) for which the CPDAG is insufficient to identify the causal effect – our distances capture these cases that cannot be identified in Gguess and compare whether this agrees with those not identifiable in Gtrue. Our motivation is to assess the object identifiable from data (oftentimes only the CPDAG) and consider it unrealistic that a practitioner would choose a random DAG in the equivalence class to derive identification formulas for causal effects that are plain wrong for other members in the equivalence class.
> We hope this perspective helps to motivate our distances and why they are useful also for CPDAGs. We will elaborate on this motivating perspective in the introduction, in a paragraph following Definition 5, and discuss the non-identifiable CPDAG-case in Section 5.1 (see also detailed comment below).
>
> ---
>
> (We respond to your detailed comments in a second reply.)
>
> (We are unsure about the visibility of messages on openreview; in case our second message is not visible to you we post a __shortened__ answer to your Q5 comments below.)
>
> Thanks for a careful reading – typos fixed, reference included, implemented minor clarifications.
>
> Re some remaining questions:
>
> * Multiple causal orders can be compatible with a given graph (DAGs induce only a partial order) and a transposition-count-based distance will need to account for this multiplicity. One would need to prove that such a distance admits an interpretation in terms of correct effect identification (cf. Sec 5.2).
>
> * Prop 7 is specific to DAGs (cf. paragraph below Prop 7). You ask how the distance between two DAGs relates to that between the corresponding CPDAGs: we did not find a relation and conjecture a theoretical characterization may be tricky: We will add an example to Section 5.1.
>
> * Yes, any two CPDAGs with no identifiable effect have distance zero. If no effect is identifiable in CPDAG Gtrue and a discovery algorithm is used that can only identify the graph up to its equivalence class (e.g. FGES), this is the correct thing to do.
>
> * Our framework makes explicit what is needed to extend to more general graphs (e.g. the graphs returned by PC without conflict-resolution): a sound and complete identifier/verifier; our distances are thus only implemented and computable for DAGs/CPDAGs. Essentially, it is unclear how one would use a graph with conflicts to infer all effects.

---

### Official Review · Reviewer_zVAE · 2024-03-13

**Q2-1 Originality-Novelty:** 2
**Q2-2 Correctness-Technical Quality:** 3
**Q2-5 Clarity Of Writing:** 3

**Q1 Summary And Contributions:**

This paper introduces a new framework for defining distances between graphs called the Adjustment Identification Distance (AID), which can be seen as a generalization of the SID by Peters & Bühlmann (2015). The main difference is that the SID constructs the distance based on parent adjustment, while the AID can accommodate other adjustment sets (specifically, they consider ancestor adjustment and optimal adjustment). In addition, the AID is more computationally efficient than the SID, which is useful for e.g. simulation studies in causal discovery.

**Q2-3 Extent To Which Claims Are Supported By Evidence:**

3: Good: the main claims are supported by convincing evidence (in the form of adequate experimental evaluation, proofs, (pseudo-)code, references, assumptions).

**Q2-4 Reproducibility:**

3: Good: key resources (e.g. proofs, code, data) are available and key details (e.g. proofs, experimental setup) are sufficiently well-described for competent researchers to confidently reproduce the main results.

**Q3 Main Strengths:**

(1) Generality: The AID accommodates different adjustment/identification methods.

(2) Faster runtime: Computing the AID is faster than e.g. the SID, making it more useful for evaluating causal discovery algorithms in simulation studies.

**Q4 Main Weakness:**

(1) Novelty: The AID is a generalization of the SID, and in that sense this paper does not contribute with particularly novel ideas. The authors mention that the method accommodates other identification methods, and it would have been interesting to see examples other than covariate adjustment.

(2) Usefulness: While the fast runtime is highly useful, I'm not convinced that the information obtained from this method is useful. I would have liked the authors to argue more strongly why these distances are interesting. It seems like using a specific type of adjustment set for computing the distance is mainly useful if one is interested in identifying causal effects using that particular type of adjustment set. However, if I understand this correctly, the verifier only checks whether the given adjustment set is a valid adjustment set, and not whether it actually is a set of ancestors/parents/optimal nodes in the true graph. This means that the distance might be 0 even if none of the recovered adjustment sets actually include ancestors/parents/optimal nodes.

(3) Motivation: It is not clear to me what the authors want the distances to reflect. I would expect that the distances are meant to reflect how well causal effects are identified, but in Lemma 10 and Example 18 the authors criticize the distances based on parent and optimal adjustment sets for not reflecting the fact that the estimated graphs respect the causal order of the true graphs (since the distances are not 0). In my opinion, this misses the point since the true and estimated graphs in these examples are quite different, and you would need quite different adjustment sets for identifying causal effects. If the authors believe that the distance should reflect the causal order, then it seems like ancestor adjustment is the only interesting adjustment set.

**Q5 Detailed Comments To The Authors:**

(1) I’m curious how the method performs on estimated graphs that are not necessarily DAGs or CPDAGs. In practice (e.g. for some versions of the PC algorithm) the estimated graph does not necessarily represent an equivalence class due to e.g. conflicts.

(2) It might be interesting to discuss how much information is lost by not considering all DAGs in the equivalence class (compared to the SID).

(3) Typos:
- “lear” should be “learn”  (first line in section 4.2).
- In the sentence “The number of edges that differ between graphs not a performance metric for causal discovery” (page 1) it seems like a word is missing.

**Q9 Complying With Reviewing Instructions:**

Yes

---

> ### Author Rebuttal · Authors · 2024-04-04
>
> Dear Reviewer zVAE,
>
> Thank you for the overall positive feedback and constructive comments. In your strength assessment you mention that our framework is very general and that we provide a fast implementation. We would like to emphasize that our implementation is not merely fast but is in fact the first implementation of a causal CPDAG-to-CPDAG distance that is computationally tractable (runtime polynomial in input size rather than exponential) and also to our knowledge the first implementation of a causal distance that remains feasible for >500 node DAGs (polynomial runtime complexity of one lesser degree than SID). To achieve these advances both our new verifier-identifier framework and new technical algorithmic contributions were required (see Section 6 & Appendix D). We therefore believe that AID does contribute substantial and novel ideas beyond the SID (__Q4(1)__).
>
> You raise the question what precisely the adjustment distances measure to motivate why they are interesting. We thank you for observing that the motivation was brief in the submission and agree that it is worth elaborating on (we will use the additional space in the camera-ready version for this).
> Our perspective is that a strategy-specific distance measures how well a guess graph (learnt via causal discovery or guessed by subject experts) performs as a tool for causal inference when used by a hypothetical practitioner to infer a causal effect. To model the practitioner we assume that they employ a principled approach to inferring causal effects given a guess of the causal graph, that is, an identification strategy. If AID(Gtrue, Gguess)=0, then for any T and Y the practitioner would, using the given identification strategy, identify the correct causal effect for T on Y using Gguess instead of Gtrue; here Gguess indeed is a great guess of Gtrue even if it was not identical to Gtrue as it yields all the right answers for the effect of any T on any Y (__Q4(2)__).
> We think each of the three adjustment identification distances we propose has their merit depending on how one believes a hypothetical practitioner would likely act, that is, how the causal knowledge encoded by the graph would be used. For example, the parent distance is natural because adjusting for the causes of treatment is already popular but the parent distance also implicitly assumes that a practitioner would try to estimate a causal effect via parent-adjustment even if Y is a non-descendant of X. The former is a plausible assumption; the latter less so in our opinion. The ancestor strategy is natural if we assume that practitioners generally believe that causal discovery algorithms learn the causal order better than the overall graph. Finally, the optimal strategy is natural if we believe that practitioners are likely to try and select adjustment sets that are not just valid but also efficient. This is why we propose three distances but leave it to the user to decide which (combination) they prefer (__Q4(3)__). We hope this perspective helps to motivate our distances and why they are useful. We will elaborate on this motivating perspective in the introduction, in a paragraph following Definition 5, and expand the introduction to Section 4 to communicate this practitioner oriented perspective more clearly to the reader and to better highlight the considerations in choosing among the distances which is currently discussed in Sections 4.1, 4.2, and 4.3.
>
> One strength of our framework is, that it make explicit what theoretical advances are necessary to extend the distances to more general graphs (e.g. the graphs returned by PC without conflict-resolution): a sound and complete identification and verification strategy; the implemented identifier/verifier we use are sound and complete for DAGs/CPDAGs and our distances are thus only implemented for DAGs/CPDAGs (__Q5(1)__).
> Thanks for raising the question what information AID looses when comparing CPDAGs as opposed to SID (which cycles through the equivalence class). This question is difficult to answer for us as it is not clear to us what the SID between CPDAGs is trying to measure: We believe that it is implausible that a practitioner would perform causal inference on a random equivalence class member in settings where it is known that only the equivalence class can be learnt. In contrast, our proposed distance reflects how many causal effects would be wrongly inferred when using the guessed CPDAG for causal inference directly (__Q5(2)__). Note that the CPDAG-to-CPDAG SID does not scale to more than a dozen of nodes at which point it ceases to cycle through all DAGs in the equivalence class and uses a crude approximation without theoretical guarantees and thus necessarily looses information.
> Finally, we thank you for pointing out two typos (fixed; __Q5(3)__).

---

### Official Review · Reviewer_gMVk · 2024-03-18

**Q2-1 Originality-Novelty:** 3
**Q2-2 Correctness-Technical Quality:** 3
**Q2-5 Clarity Of Writing:** 4

**Q1 Summary And Contributions:**

The paper proposes a framework for developing causal distances between graphs, while also providing specific examples of such distances for adjustment-based identification formulas. One of the proposed distances, the parent adjustment distance, is shown to be the same as the SID. The next sections describe improvements over this distance using the ancestral and optimal adjustment sets respectively. These results are then also extended to CPDAGS. These are the first causal distances between CPDAGs with a polynomial runtime guarantee. The claimed time complexity of being quadratic in the number of nodes of the graph is also validated empirically, and it is also shown that the proposed implementation is faster and scales more easily compared to existing methods for calculating SID.

**Q2-3 Extent To Which Claims Are Supported By Evidence:**

3: Good: the main claims are supported by convincing evidence (in the form of adequate experimental evaluation, proofs, (pseudo-)code, references, assumptions).

**Q2-4 Reproducibility:**

4: Excellent: key resources (e.g. proofs, code, data) are available and key details (e.g. proof sketches, experimental setup) are comprehensively described for competent researchers to confidently and easily reproduce the main results.

**Q3 Main Strengths:**

* The writing and flow are very nice, and the paper is easy to read.
* The work is contextualized very well in existing literature.
* The proposed framework as well as the distances are simple and elegant, but quite impactful.
* It's very good that the proposed distances have been provided as a package.
* The topic is timely and relevant. Any proposal which helps to improve in some form the robustness of causal discovery evaluations is very welcome.

**Q4 Main Weakness:**

* Given the arguments made in the introduction, it would have been nice to see how the proposed distances compare with each other in an actual causal discovery task. It's one thing to see counterexamples where they would differ in logical ways but if in most cases they're all the same then I guess it wouldn't be as useful despite being interesting.
* It is quite unclear how the existence of a verifier would extend to identification strategies that go beyond adjustment sets.

**Q5 Detailed Comments To The Authors:**

**Questions**
1. How easy or difficult do you think it would be to have verifiers for identification methods that are not adjustment set based?
2. Could the framework be extended to incorporate further assumptions about the data-generating process as well?

**Comments**
1. Section 4.2 first line: 'lear' -> learn
2. 'Nonetheless, the Parent-AID between two DAGs with the same causal orders can be large.' -> Is this line in the wrong place by any chance?
3. Section 7 1st paragraph: 'ompatible' -> compatible

**Q9 Complying With Reviewing Instructions:**

Yes

---

> ### Author Rebuttal · Authors · 2024-04-04
>
> Dear Reviewer gMVk,
>
> Thank you for the very positive feedback. We are particularly happy that you agree that our framework is simple yet elegant and impactful. In your review you also raise three interesting points.
>
> __First,__ it would be good to verify whether and by how much the different distances differ in practice, particularly in an actual causal discovery task.
>
> We agree and we will add two additional simulation studies in the camera-ready version. The first is a study in which we compute the average distances (parent-AID, ancestor-AID, oset-AID, SHD) between two random Erdős–Rényi graphs for a variety of parameter settings. Please see our general response for results of this experiment on dense graphs (we will run and include more settings with different graph sizes and densities for the camera-ready version). This average distance can be thought of as a random guessing performance baseline for the respective distance and is therefore a useful guideline as to what a low or high distance is. We will also include a similar experiment where we compute the distances between a candidate graph that is missing a single edge when compared to the true graph, to give further intuition. We compute the correlation between our respective distances in both settings. Our correlation results (please see the general response for the correlation tables) indicate that each of the distances we consider captures distinct information. We agree that a simulation study that investigates how different causal discovery algorithms perform in terms of those metrics is an interesting avenue for future research but consider it beyond the scope of this paper, in which we develop and contribute a general framework and new algorithms for causal graph distances.
>
> __Second__, the question of how a more general verifier could be used to define alternative strategy-specific distances.
>
> Verification has not really received any research attention so far and as a result the only available verifiers are necessary and sufficient identification results such as the valid adjustment criterion or the instrumental variable criterion (cf. our discussion in Section 3). In principle, we could use the latter to propose an instrumental variable based distance. However, IV is generally not complete for identification, that is, there may be identifiable causal effects that are not identifiable with an instrumental variable approach (consider for example any DAG on two nodes such as X → Y), which is why we did not pursue this avenue (in our framework we build on sound and complete identification strategies, cf. Section 3.1). In general, the problem of verification is not too dissimilar to causal effect identification. In identification we try to find an identifying formula, e.g., by applying the do-calculus. In verification we are given a candidate formula and try to verify whether it is an identifying formula. It seems plausible that this should be feasible, for example, by employing the do-calculus, particularly if we restrict ourselves to a certain class of identifying formula (e.g. the g-formula or the front-door approach). Given a verifier suitable for the inputs provided by some candidate identification strategy we can then follow our framework to define a distance like we did in the case of the parent-, ancestor- or oset-distances.
>
> __Third__, could the framework be extended to incorporate further assumptions about the data-generating process as well?
>
> The distances in our framework are entirely graphical in the sense that they only take into account the two graphs $G_{\mathrm{guess}}$ and $G_{\mathrm{guess}}$. We are unsure about the question to incorporate assumptions on the data generating process: it seems new distances that somehow take into account properties beyond the graph would then be distances between structural equation models or another class of causal models rather than between causal graphs; this may be an interesting yet separate avenue for future research.
>
>
> __Finally__, we have also fixed the typos you have pointed out to us and would like to thank you again for the constructive feedback.

---

### Official Review · Reviewer_CXNH · 2024-03-21

**Q2-1 Originality-Novelty:** 3
**Q2-2 Correctness-Technical Quality:** 3
**Q2-5 Clarity Of Writing:** 3

**Q1 Summary And Contributions:**

This work provides a general framework of CPDAG distance metrics based on adjustment, aiming to improve sensitivity over structural distance, with the later as a special case. Algorithms and complexity are also provided.

**Q2-3 Extent To Which Claims Are Supported By Evidence:**

3: Good: the main claims are supported by convincing evidence (in the form of adequate experimental evaluation, proofs, (pseudo-)code, references, assumptions).

**Q2-4 Reproducibility:**

3: Good: key resources (e.g. proofs, code, data) are available and key details (e.g. proofs, experimental setup) are sufficiently well-described for competent researchers to confidently reproduce the main results.

**Q3 Main Strengths:**

This work fills a gap for DAG learning performance metrics, for which subject DAG-based metrics are often used to compare CPDAGs, and is unfair/not sensitive in practical scenarios. The proposed framework is sound and still interpretable, and simulation shows reasonably fast computation. The example where SID can be huge for "good" graphs is very helpful.

**Q4 Main Weakness:**

the flipside of the more general adjustment based framework is that the metrics becomes harder to interpret compared to SHD/SID, especially relevant to graph size p. The submission could have provided more help on interpretations and numerical examples.

**Q5 Detailed Comments To The Authors:**

See major comment above: since the metrics are eventually mostly used for numerical comparison between algorithms, and in most cases as a function of graph size/density, it would really benefit the popularization of proposed metrics if authors could provide some comparisons in such scenarios.

**Q9 Complying With Reviewing Instructions:**

Yes

---

> ### Author Rebuttal · Authors · 2024-04-04
>
> Dear Reviewer CXNH,
>
> Thank you the assessment that our framework fills an important gap in performance metrics for graph learning by providing a conceptually sound causal distance between CPDAGs.
>
> You also raise the point that it would be helpful if we included more help on how to interpret our distances both via explanations and by including more numerical examples. We agree that this is worth elaborating on (we will use the additional space in the camera-ready version for this):
> To give practitioners more intuition as to whether a specific distance value is low or high we will include two additional simulation studies in the camera-ready version. The first is a study in which we compute the average distances (parent-AID, ancestor-AID, oset-AID, SHD) between two random Erdős–Rényi graphs for a variety of parameter settings. Please see our general response for results of this experiment on dense graphs (we will run and include more settings with different graph sizes and densities for the camera-ready version). This average distance can be thought of as a random guessing performance baseline for the respective distance and is therefore a useful guideline as to what a low or high distance is. We will also include a similar experiment where we compute the distances between a candidate graph that is missing a single edge when compared to the true graph, to give further intuition.
> We believe the inclusion of these additional results improves the paper and hope our response addresses your concern. Furthermore, our novel algorithms enable efficient on-demand calculation of such reference values for any graph distribution used in a given benchmarking study by averaging distances quickly calculated over hundreds of graph-pairs randomly drawn from this distribution.
>
> We are unsure if we correctly understand your comment regarding AID becoming harder to interpret than SHD/SID with respect to graph size $p$. We propose to report the normalized distance between 0 and 1 as the fraction of causal effects that would be wrongly inferred when using Gguess in place of Gtrue, which amounts to dividing the number of mistakes by $p(p-1)$; the unnormalized number of (T,Y)-pairs for which the identification based on Gguess is wrong (this mistake count is an integer between 0 and $p(p-1)$) is returned by our implementation, too (and can be obtained from the normalized distance by multiplying by $p(p-1)$).

---

### Official Review · Reviewer_rwGx · 2024-03-23

**Q2-1 Originality-Novelty:** 2
**Q2-2 Correctness-Technical Quality:** 3
**Q2-5 Clarity Of Writing:** 3

**Q1 Summary And Contributions:**

The authors propose new framework for measuring the distance between graphical structures representing causal models. The aim was to develop a new tool that would better take into account causality / identification aspects than the popular ones such as structural Hamming distance (SHD). The authors also analyze the computational efficiency of the proposed methods and provide Rust implementations with
a Python interface.

**Q2-3 Extent To Which Claims Are Supported By Evidence:**

3: Good: the main claims are supported by convincing evidence (in the form of adequate experimental evaluation, proofs, (pseudo-)code, references, assumptions).

**Q2-4 Reproducibility:**

4: Excellent: key resources (e.g. proofs, code, data) are available and key details (e.g. proof sketches, experimental setup) are comprehensively described for competent researchers to confidently and easily reproduce the main results.

**Q3 Main Strengths:**

- Overall, this is a well-written and interesting paper.

- The authors provide an implementation of the proposed framework that may be useful for researchers.

- Experimental results quite convincingly explain runtimes of the implemented algorithms.

**Q4 Main Weakness:**

- The contribution of this work is not particularly novel. Seems like a natural generalization of the Structural Intervention Distance (SID).

- It would be interesting to see some experimental evidence confirming the usefulness of the new framework, e.g. in the context of causal structure learning.

**Q5 Detailed Comments To The Authors:**

See my comment above.

**Q9 Complying With Reviewing Instructions:**

Yes

---

> ### Author Rebuttal · Authors · 2024-04-04
>
> Dear Reviewer rwGx,
>
> Thank you for the assessment that the paper is interesting, convincingly establishes our algorithms' excellent runtimes, and that our framework and implementation may be useful for researchers.
>
> In your review you raise two main concerns:
>
> First, regarding the novelty of our contribution when compared to the original SID paper. We would like to emphasize that our implementation is not merely fast but is in fact the first implementation of a causal CPDAG-to-CPDAG distance that is computationally tractable (runtime polynomial in input size rather than exponential) and also to our knowledge the first implementation of a causal distance that remains feasible for >500 node DAGs (polynomial runtime complexity of one lesser degree than SID). To achieve these advances both our new verifier-identifier framework and new technical algorithmic contributions were required (see Section 6 & Appendix D). We therefore believe that AID does contribute substantial and novel ideas beyond the SID.
>
> Second, you raise the point that more experimental evidence to establish the usefulness of our distances would be helpful. We agree that characterizing how and when the distances differ is an interesting avenue for future research and we likewise hope that future benchmark studies of causal discovery algorithms include these additional distances to present a more complete picture of the relative strengths of the various algorithms. A full comparison is beyond the scope of the present submission.
>
> We do agree, that illustrating empirically that the distances differ, is a valuable addition that will improve the paper (we will use the additional space in the camera-ready version for including the below). We ran two experiments to compare the distances:
>
> __Experiment A__: We draw 300 pairs (Gtrue, Gguess) of random dense graphs (30 nodes, edge probability 0.3, on average 130.5 edges). The distances calculated on those 300 graph-pairs differ and we obtain the following correlation matrix:
>
> |              |   ancestor_aid |   oset_aid |   parent_aid |      shd |
> |:-------------|---------------:|-----------:|-------------:|---------:|
> | ancestor_aid |       1        |   0.9474   |     0.7429   | 0.4673   |
> | oset_aid     |       0.9474   |   1        |     0.5715   | 0.5232   |
> | parent_aid   |       0.7429   |   0.5715   |     1        | 0.1769   |
> | shd          |       0.4673   |   0.5232   |     0.1769   | 1        |
>
> and mean values:
>
> |              |  mean distance |
> |:-------------|---------------:|
> | ancestor_aid |       0.5830   |
> | oset_aid     |       0.5948   |
> | parent_aid   |       0.8805   |
> | shd          |       0.4652   |
>
> __Experiment B__: We draw 300 pairs (Gtrue, Gguess) of graphs, where Gtrue is a random dense graph (30 nodes, edge probability 0.3, on average 130.5 edges) and Gguess is Gtrue with one edge removed at random. The SHD(Gtrue, Gguess)=1 for all those pairs. The adjustment identification distances calculated on those 300 graph-pairs differ and we obtain the following correlation matrix:
>
> |              |   ancestor_aid |   oset_aid |   parent_aid |
> |:-------------|---------------:|-----------:|-------------:|
> | ancestor_aid |      1         |   0.7281   |    0.0886    |
> | oset_aid     |      0.7281    |   1        |    0.2080    |
> | parent_aid   |      0.0886    |   0.2080   |    1         |
>
> and mean values:
>
> |              |  mean distance |
> |:-------------|---------------:|
> | ancestor_aid |       0.0045   |
> | oset_aid     |       0.0136   |
> | parent_aid   |       0.0257   |
> | shd          |       0.0023   |
>
> __The above results__, and additional settings plus scatter plots which we will include in the appendix of the camera-ready version, illustrate that the distances measure distinct information. Average distances can be efficiently calculated over several hundreds of graph-pairs to establish a reference value for any given graph distribution used in a causal discovery benchmarking study.

---

### Meta-Review · Area_Chair_2cf8 · 2024-04-17

The paper proposes a new metric for causal graph evaluation called the Adjustment Identification Distance (AID). It is a twist on the Structural Intervention Distance, that focusses on causal effect estimation rather than structural/graphical properties captured by the familiar Structural Hamming Distance (SID), but allows for more general adjustment criteria than SID, and is faster / more efficient to compute

Strengths:
- well-written paper, easy to read
- the proposed AID metric is more general and faster/easier to compute than the related SID metric

Weakness:
- interpretation of the new metric is likely to become even more challenging to non-experts in causality
- relies on causal sufficiency (DAG/CPDAG)
- limited added novelty compared to SID (vs. SHD)
- runtime to compute the causal graph quality metric may be of limited importance in practice

Reviewers were all positive, with scores ranging from borderline to strong accept, although with some concerns on interpretability.

I am happy to follow the general consensus among reviewers. However, on reading the paper I largely agree with the more cautious assessment of reviewer zVAE. It is a nice addition to the conference, and it will find an interested audience, but it does not provide any truly novel insights, comes with strong limitations (no latent confounders), and I am concerned that with yet another causal metric (even though it comes with a nice ‘easy to compute’ bonus)  it will become even harder for practitioners to make sense of it all. In short: ok contribution, clearly good enough to recommend accept, but no eye-opener …